# FASTERCACHE: TRAINING-FREE VIDEO DIFFUSION MODEL ACCELERATION WITH HIGH QUALITY

**Zhengyao Lv**[1*]   **Chenyang Si**[2‡]   **Junhao Song**[3]   **Zhenyu Yang**[3]
**Yu Qiao**[3]   **Ziwei Liu**[2†]   **Kwan-Yee K. Wong**[1†]
[1]The University of Hong Kong   [2]S-Lab, Nanyang Technological University
[3]Shanghai Artificial Intelligence Laboratory
**Code: https://github.com/Vchitect/FasterCache**

## ABSTRACT

In this paper, we present *FasterCache*, a novel training-free strategy designed to accelerate the inference of video diffusion models with high-quality generation. By analyzing existing cache-based methods, we observe that *directly reusing adjacent-step features degrades video quality due to the loss of subtle variations*. We further perform a pioneering investigation of the acceleration potential of classifier-free guidance (CFG) and reveal significant redundancy between conditional and unconditional features within the same timestep. Capitalizing on these observations, we introduce FasterCache to substantially accelerate diffusion-based video generation. Our key contributions include a dynamic feature reuse strategy that preserves both feature distinction and temporal continuity, and CFG-Cache which optimizes the reuse of conditional and unconditional outputs to further enhance inference speed without compromising video quality. We empirically evaluate FasterCache on recent video diffusion models. Experimental results show that FasterCache can significantly accelerate video generation (*e.g.*, 1.67× speedup on Vchitect-2.0) while keeping video quality comparable to the baseline, and consistently outperform existing methods in both inference speed and video quality.

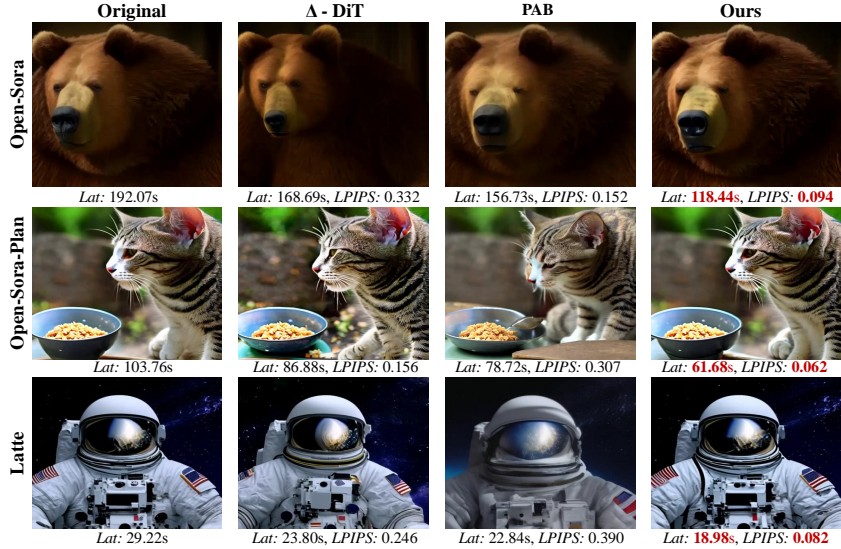

(Lat denotes latency, measured on a single A100 GPU. Video synthesis configuration: 192 frames at 480P for Open-Sora, 65 frames at 512×512 for Open-Sora-Plan, and 16 frames at 512×512 for Latte.)

Figure 1: Comparison of visual quality and inference speed with competing methods.

## 1 INTRODUCTION

Diffusion transformers (DiT) (Peebles & Xie, 2023) have achieved notable success in image (Chen et al., 2023; 2024b; Esser et al., 2024) and video generation (Ma et al., 2024a; Zheng et al., 2024; PKU-Yuan Lab and Tuzhan AI etc., 2024), attracting significant attention for their potential. Although iterative denoising, classifier-free guidance (CFG) (Ho & Salimans, 2022), and transformer

---

† Corresponding authors. ‡ Project leader. *The work was done during an internship at Shanghai AI Lab.

Figure 2: Vanilla cache-based acceleration method typically reuses features cached from previous timesteps directly for the current timestep.

attention mechanisms have significantly improved the generative capabilities of diffusion models, they also lead to substantial computational costs and increased memory requirements for inference, especially for video generation which typically takes 2-5 minutes to synthesize a 6-second 480P video, limiting their practical use. This calls for the development of new techniques that require less computational cost for diffusion models (Salimans & Ho, 2022; Ma et al., 2024b; Chen et al., 2024c; Zhao et al., 2024c).

Among the recently proposed solutions, cache-based acceleration has emerged as one of the most widely adopted approaches. This approach speeds up the sampling process by reusing intermediate features across timesteps, thereby reducing redundant computations and significantly improving computational efficiency. Besides, it requires no additional training costs for inference acceleration and offers straightforward generalization to other video diffusion models. Examples include cache-based methods for U-Net based diffusion models (Ma et al., 2024b; Li et al., 2023b), residual caching in $\Delta$-DiT (Chen et al., 2024c) for transformer based diffusion models, and hierarchical attention caching of PAB (Zhao et al., 2024c) for video generation. Despite their proven effectiveness, there exist two critical concerns: 1) Whether directly reusing intermediate features aligns with the iterative denoising mechanism, considering the inherent feature variations between timesteps. 2) Current cache-based methods focus primarily on the attention features within the transformer networks, with limited exploration of accelerating different parts of the pipeline. In this work, we aim to address these two concerns.

To thoroughly investigate the acceleration potential of DiT inference for video generation, we delve into the feature reuse process of existing cache-based methods. As shown in Fig. 2, these methods typically assume a high degree of feature similarity between adjacent timesteps in the iterative denoising process, and achieve accelerated inference by sharing features across consecutive timesteps. However, our investigation reveals that while features in the same attention module (*e.g.*, spatial attention) appear to be nearly identical between adjacent timesteps, there exist some subtle yet discernible differences. As a result, *a naive feature caching and reuse strategy often leads to degradation of details in generated videos*, as shown in Fig. 3 (a).

Following this analysis, we further extend the scope of our investigation to explore potential redundancy within the classifier-free guidance (CFG). As shown in Fig. 3 (b), compared to internal network modules (*e.g.*, spatial attention and temporal attention), CFG almost doubles the inference time due to the additional computation required for unconditional outputs. Our experiments reveal a notable difference from our earlier conclusion regarding attention modules. In CFG, *the conditional and unconditional outputs at the same timestep exhibit a very high degree of similarity, suggesting significant information redundancy.* In contrast, the similarity of unconditional features between adjacent timesteps is relatively weak. We further discover that the differences between the conditional and unconditional outputs are predominantly concentrated in low- to mid-frequency features during the mid-sampling phase, shifting to high-frequency features in the late-sampling phase, with these differences evolving gradually.

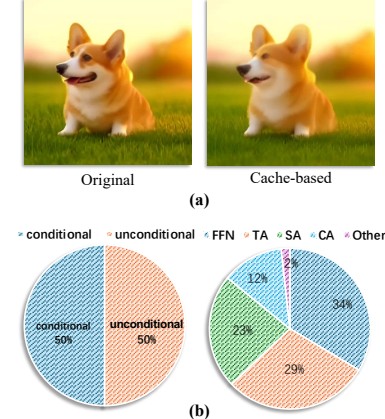

Figure 3: (a) Vanilla cache-based methods typically lead to detail loss. (b) Time overhead proportions of different components in video models.

Based on the above insights, we propose a novel strategy, termed *FasterCache*, to accelerate the inference of video diffusion models while ensuring high-quality generation and remaining training-free. Specifically, we first introduce a dynamic feature reuse strategy for attention modules which dynamically adjusts the reused features across different timesteps, ensuring both distinction and

continuity of features between adjacent timesteps are maintained. This strategy preserves the subtle variations essential for the iterative denoising process while ensuring temporal consistency, resulting in accelerated inference with minimal loss of details in the generated videos. Furthermore, we introduce CFG-Cache, an innovative technique that stores the residuals between conditional and unconditional outputs, dynamically enhancing their high-frequency and low-frequency components before reuse. This significantly accelerates inference while preserving details in generated videos.

We evaluate our FasterCache on various video diffusion models, including Open-Sora 1.2 (Zheng et al., 2024), Open-Sora-Plan (PKU-Yuan Lab and Tuzhan AI etc., 2024), Latte (Ma et al., 2024a), CogVideoX (Yang et al., 2024), and Vchitect-2.0 (Fan et al., 2025). As shown in Fig 1, experimental results demonstrate that FasterCache can significantly accelerate inference while preserving high-quality video generation across all tested models. Specifically, on Vchitect-2.0, FasterCache achieves $1.67\times$ speedup, with performance comparable to the baseline (VBench: baseline 80.80% $\rightarrow$ FasterCache 80.84%). Furthermore, our method outperforms existing approaches in both inference speed and video generation quality, highlighting its effectiveness and efficiency in real-world applications.

Overall, the contributions of this work are as follows:

- We analyze the feature reuse process in cache-based methods and discover that while adjacent-step features in attention modules appear to be similar, their subtle differences can degrade output quality if ignored.
- We conduct a pioneering investigation of CFG's potential for acceleration, finding high redundancy within the same timestep but weaker similarity across adjacent timesteps, revealing new acceleration opportunities.
- We propose *FasterCache*, a training-free strategy that dynamically adjusts feature reuse, preserving both feature distinction and continuity. It also introduces CFG-Cache to accelerate inference while preserving details in generated videos.
- We empirically evaluate our approach on various video diffusion models, demonstrating significant improvement in inference speed while maintaining high video quality.

## 2 METHODOLOGY

### 2.1 PRELIMINARY

**Diffusion model** is a generative model consisting of a forward process and a reverse process. Specifically, its forward diffusion process progressively adds noise to the data $\boldsymbol{x}_0 \sim p_{data}(\boldsymbol{x}_0)$, eventually destroying the signal. This can be formulated as:

$$q(\boldsymbol{x}_t|\boldsymbol{x}_0) = \mathcal{N}(\boldsymbol{x}_t; \sqrt{\alpha_t}\boldsymbol{x}_0, \sqrt{1-\alpha_t}\boldsymbol{I}), \tag{1}$$

where $\{\alpha_t\}_{t=1}^T$ controls the noise schedules and T represents the total number of diffusion timesteps. The reverse process is typically parameterized as a UNet or transformer architecture $\boldsymbol{\epsilon}_\theta$ which is trained to predict the noise with the following loss function:

$$\mathcal{L}_{DM} = \mathbb{E}_{\boldsymbol{x}, \boldsymbol{\epsilon} \sim \mathcal{N}(0,1), t}[||\boldsymbol{\epsilon} - \boldsymbol{\epsilon}_\theta(\boldsymbol{x}_t, t)||_2^2]. \tag{2}$$

A clean signal $\boldsymbol{x}_0$ can be recovered through iterative inference steps which predict $\boldsymbol{x}_{t-1}$ from $\boldsymbol{x}_t$ using $\boldsymbol{\epsilon}_\theta$. This can formulated as:

$$p(\boldsymbol{x}_{t-1}|\boldsymbol{x}_t) = \mathcal{N}(\boldsymbol{x}_{t-1}; \mu_\theta(\boldsymbol{x}_t, t), \Sigma_\theta(\boldsymbol{x}_t, t)), \tag{3}$$

where $\mu_\theta$ and $\Sigma_\theta$ are the mean and variance parameterized with learnable $\theta$.

**Video diffusion models** recently employ diffusion transformers as the backbone for noise prediction. This work explores video synthesis acceleration based on Open-Sora 1.2 (Zheng et al., 2024). This model is composed of 56 stacked transformer layers, with alternating spatial and temporal layers. Each layer contains not only a spatial or temporal attention module but also a cross-attention and a feed-forward network. Latte (Ma et al., 2024a) and Open-Sora-Plan (PKU-Yuan Lab and Tuzhan AI etc., 2024) also adopt a similar architecture as their noise prediction networks.

**Classifier-Free Guidance (CFG)** has proven to be a powerful technique for enhancing the quality of synthesized images/videos in diffusion models. During the sampling process, CFG computes two

outputs, namely $\epsilon_\theta(\boldsymbol{x}_t, \boldsymbol{c})$ for the conditional input $\boldsymbol{c}$ and $\epsilon_\theta(\boldsymbol{z}_t, \emptyset)$ for the unconditional input $\emptyset$ (often an empty or negative prompt). The final output is given by:

$$\tilde{\epsilon}_\theta(\boldsymbol{x}_t, \boldsymbol{c}) = (1+g)\epsilon_\theta(\boldsymbol{x}_t, \boldsymbol{c}) - g\epsilon_\theta(\boldsymbol{z}_t, \emptyset), \tag{4}$$

where $g$ is the guidance scale. As shown in Fig. 3 (b), while CFG significantly enhances visual quality, it also increases computational cost and inference latency due to the additional computation required for unconditional outputs.

## 2.2 RETHINKING ATTENTION FEATURE REUSE

Attention feature reuse has become a primary focus for cache-based acceleration methods in video generation (*e.g.*, pyramid attention reuse of PAB). In video diffusion models, features of attention modules (*e.g.*, spatial attention and temporal attention) exhibit a high similarity between adjacent timesteps, as illustrated in Fig. 4. Hence, existing methods completely bypass the attention computations in subsequent timesteps by reusing the cached attention features, thereby significantly reducing computational costs.

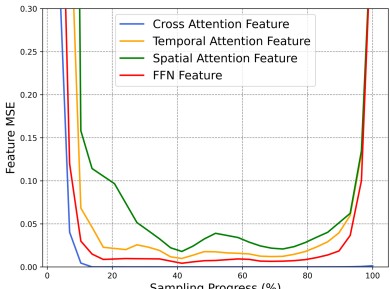

Figure 4: Comparison of the mean squared error (MSE) of attention features between the current and previous diffusion steps. Smaller values indicate higher similarity.

To gain a better understanding of the implications of attention feature reuse in video generation, we first visualize the videos generated with the same random seed and observe that existing feature reuse methods result in a noticeable loss of details in the output. For example, as illustrated in Fig. 5, compared to the original video generated without feature reuse, the video generated with vanilla feature reuse exhibits a smoother sky, with a lack of visible stars, indicating a noticeable degradation in fine details.

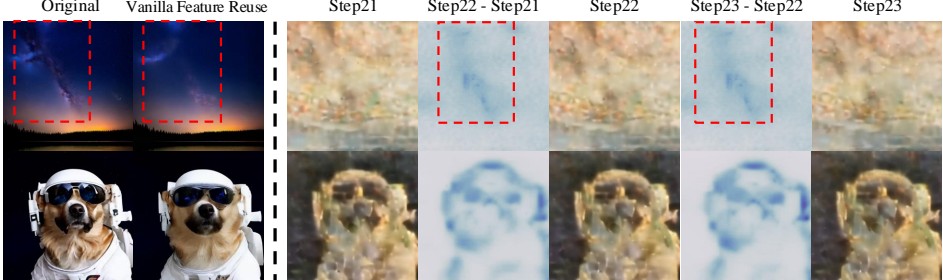

Figure 5: Visual quality degradation caused by Vanilla Feature Reuse (left) and feature differences between adjacent timesteps (right).

To investigate the underlying causes of this phenomenon, we subsequently visualize the attention features between adjacent timesteps and analyze their differences. The results indicate that while the attention features between adjacent timesteps are highly similar, there exist noticeable differences between them. These subtle variations between timesteps are essential for preserving fine details in video generation. Therefore, directly reusing features without accounting for these differences leads to the loss of important visual information, resulting in smoother and less detailed outputs. This highlights the need for a more refined approach to feature reuse, *i.e.*, one that can retain computational efficiency while preserving key inter-step variations.

## 2.3 FEATURE REDUNDANCY IN CFG

Following the observation of feature redundancy in attention modules across adjacent timesteps, we further extend our investigation into other critical components of the diffusion models. Through this broader analysis of the entire denoising process, we find that classifier-free guidance (CFG) significantly increases inference time, as it requires the computation of both conditional and unconditional outputs at every timestep. While CFG has been widely adopted for enhancing visual quality, there is little exploration to reduce its computational burden, leaving this aspect largely uncharted.

To explore potential redundancy within CFG, we first conduct a quantitative analysis of the similarity between conditional and unconditional outputs at the same timestep as well as across adjacent timesteps based on mean squared error (MSE). As shown in Fig. 6 (a), the results reveal that, in

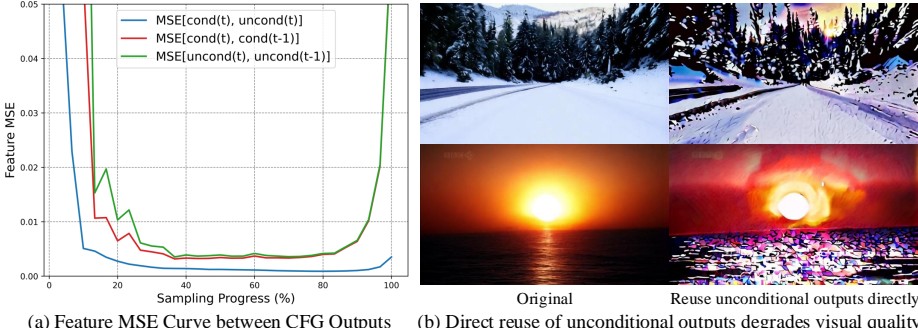

(a) Feature MSE Curve between CFG Outputs  (b) Direct reuse of unconditional outputs degrades visual quality

Figure 6: (a) The MSE between conditional and unconditional outputs at the same timestep as well as across adjacent timesteps. (b) Directly reusing unconditional outputs from previous timesteps will lead to a significantly degraded visual quality.

the mid to later stages of sampling, the similarity between conditional and unconditional outputs at the same timestep is remarkably high, significantly surpassing that of adjacent steps. Hence, as illustrated in Fig. 6 (b), directly reusing unconditional outputs from adjacent timesteps, as suggested in existing methods, leads to significant error accumulation, resulting in a decline in video quality. These results indicate substantial redundancy in the CFG process and highlight the necessity for a new strategy to accelerate CFG without compromising the quality of the generated outputs.

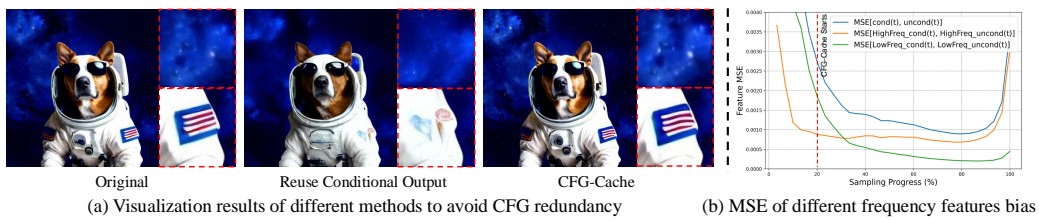

(a) Visualization results of different methods to avoid CFG redundancy  (b) MSE of different frequency features bias

Figure 7: (a) Simply reusing the conditional output from the same time step results in the poor generation of intricate details. (b) Trend curves of high and low-frequency biases between conditional and unconditional outputs change as sampling progresses.

## 2.4 FASTERCACHE FOR VIDEO DIFFUSION MODEL

Capitalizing on the above discoveries, we introduce an innovative approach, *FasterCache*, which accelerates inference for video diffusion models while preserving high-quality generation. This is accomplished through a ***Dynamic Feature Reuse Strategy*** that maintains feature distinction and temporal continuity. Furthermore, we introduce ***CFG-Cache*** to optimize the reuse of conditional and unconditional outputs, further enhancing inference speed without compromising visual quality.

**Dynamic Feature Reuse Strategy**   As discussed in Section 2.2, vanilla attention feature reuse strategy neglects the feature differences between adjacent timesteps which leads to visual quality degradation. Hence, instead of directly reusing previously cached features at the current timestep, we propose a Dynamic Feature Reuse Strategy that can more effectively capture and preserve critical details in the generated videos. Specifically, for the attention modules in diffusion models, we compute the attention module outputs at every alternate timestep. For example, we calculate the attention outputs for each layer at $t+2$ and $t$ timesteps, denoted as $\boldsymbol{F}_{t+2}$ and $\boldsymbol{F}_t$, and store them in the feature cache as $\boldsymbol{F}_{cache}^{t+2}$ and $\boldsymbol{F}_{cache}^t$. To dynamically adjust feature reuse, we compute the difference between the adjacent cached features. This serves as a bias for approximating the feature variation trend and enables the reused features to more accurately capture the evolving details across timesteps. For the intermediate $t-1$ timestep, its features can be computed as:

$$\boldsymbol{F}_{t-1} = \boldsymbol{F}_{cache}^t + (\boldsymbol{F}_{cache}^t - \boldsymbol{F}_{cache}^{t+2}) * w(t), \qquad (5)$$

where $w(t)$ is a weighting function that modulates the contribution of the feature difference to account for variation between adjacent timesteps, ensuring both efficiency and the preservation of fine details in the generated videos. In our experiments, $w(t)$ gradually increases as the sampling process progresses, allowing the model to place greater emphasis on the feature differences at later stages of generation. Further discussions on the design of feature bias term and the selection of $w(t)$ in Eq. (5) can be found in Appendix A.3.1. Consequently, our approach significantly accelerates inference while preserving the visual quality of the synthesized videos.

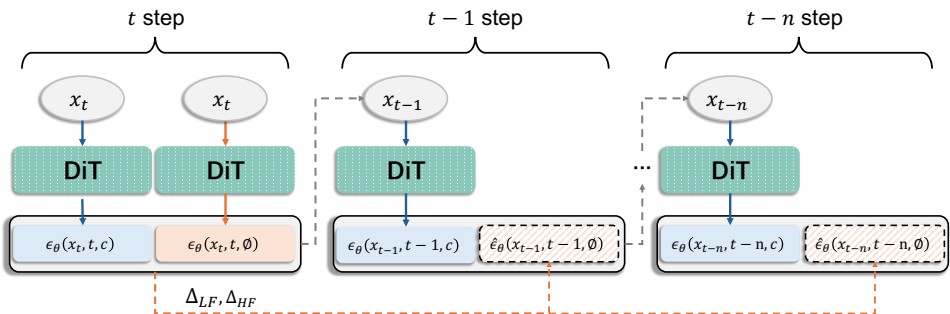

Figure 8: Overview of the CFG-Cache. CFG-Cache accelerates the computation of the unconditional output (in the dashed orange box) by caching the high- and low-frequency biases between the conditional and unconditional outputs, and dynamically enhancing them during reuse.

**CFG-Cache**    As analyzed in Section 2.3, the conditional and unconditional outputs at the same timestep exhibit high similarity in CFG, indicating significant information redundancy. A naive approach to take advantage of this would be to directly reuse the conditional features for the corresponding unconditional outputs at the same timestep. However, this often leads to a noticeable degradation in detail generation. As illustrated in Fig 7 (a), this approach results in poor generation of intricate details, such as the texture of the spacesuit which shows a lack of details and clarity. Since both the conditional and unconditional outputs in CFG represent predicted noise, and drawing inspiration from the Dynamic Feature Reuse Strategy and FreeU (Si et al., 2024), we analyze the differences between these two outputs in the frequency domain. In Fig 7 (b), we observe that, from the activation of CFG-Cache until the end of the sampling, the difference between the conditional and unconditional outputs gradually shifts from being dominated by low-frequency components to being dominated by high-frequency components. This indicates that the effects of CFG in the sampling process is primarily to influence perceptual features like layout and shape during the early and mid-stages, while contributing to detail synthesis in the later stages. A similar phenomenon can also be observed in Hsiao et al. (2024). This observation suggests that despite their overall similarity, key differences in frequency components must be addressed to avoid the degradation of fine details. More discussion and visualization can be found in Appendix A.3.2.

Building on this discovery, we propose CFG-Cache, a novel approach designed to account for both high- and low-frequency biases, coupled with a timestep-adaptive enhancement technique. Specifically, as shown in Fig. 8, at timestep $t$, a full inference is performed to obtain both the conditional output $\epsilon_\theta(x_t, t, c)$ and the unconditional output $\epsilon_\theta(x_t, t, \emptyset)$. We then separately calculate the biases for the high-frequency ($\Delta_{HF}$) and low-frequency ($\Delta_{LF}$) components between these two outputs:

$$\Delta_{LF} = \mathcal{FFT}(\boldsymbol{\epsilon}_\theta(\boldsymbol{x}_t, t, \emptyset))_{low} - \mathcal{FFT}(\boldsymbol{\epsilon}_\theta(\boldsymbol{x}_t, t, \boldsymbol{c}))_{low}, \qquad (6)$$

$$\Delta_{HF} = \mathcal{FFT}(\boldsymbol{\epsilon}_\theta(\boldsymbol{x}_t, t, \emptyset))_{high} - \mathcal{FFT}(\boldsymbol{\epsilon}_\theta(\boldsymbol{x}_t, t, \boldsymbol{c}))_{high}. \qquad (7)$$

These biases ensure that both high- and low-frequency differences are accurately captured and compensated during the reuse process. In the subsequent $n$ timesteps (from $t - 1$ to $t - n$), we infer only the outputs of the conditional branches and compute the unconditional outputs using the cached $\Delta_{HF}$ and $\Delta_{LF}$ as follows:

$$\hat{\boldsymbol{\epsilon}}_\theta(\boldsymbol{x}_{t-i}, t - i, \emptyset) = \mathcal{IFFT}(\mathcal{F}_{low}, \mathcal{F}_{high}), \qquad (8)$$

$$\mathcal{F}_{low} = \quad \Delta_{LF} * w_1 + \mathcal{FFT}(\boldsymbol{\epsilon}_\theta(\boldsymbol{x}_{t-i}, t - i, \boldsymbol{c}))_{low}, \qquad (9)$$

$$\mathcal{F}_{high} = \Delta_{HF} * w_2 + \mathcal{FFT}(\boldsymbol{\epsilon}_\theta(\boldsymbol{x}_{t-i}, t - i, \boldsymbol{c}))_{high}. \qquad (10)$$

Here, $w_1$ and $w_2$ are adaptively adjusted based on the sampling timestep $t$, with greater emphasis on different frequency components at distinct sampling phases. The weighting scheme is defined as:

$$w_1 = 1 + \alpha_1 \cdot \mathbb{I}(t > t_0), w_2 = 1 + \alpha_2 \cdot \mathbb{I}(t <= t_0), \qquad (11)$$

where $\alpha_1$ and $\alpha_2$ are hyperparameter weights, $t_0$ is the manually set switching timestep, and $\mathbb{I}(\cdot)$ is the indicator function. This formulation ensures that mid-low frequencies are prioritized in the mid-sampling phase, while high-frequency components receive more attention in the later phase.

## 3 EXPERIMENTS

### 3.1 EXPERIMENTAL SETTINGS

**Base models and compared methods**    To demonstrate the effectiveness of our method, we apply our acceleration technique to different video synthesis diffusion models, including the Open-Sora

1.2 (Zheng et al., 2024), Open-Sora-Plan (PKU-Yuan Lab and Tuzhan AI etc., 2024), Latte (Ma et al., 2024a), CogVideoX (Yang et al., 2024), and Vchitect-2.0 (Fan et al., 2025). We compare our base models with recent efficient video synthesis techniques, including PAB (Zhao et al., 2024c) and $\Delta$-DiT (Chen et al., 2024c), to highlight the benefits of our approach. Notably, $\Delta$-DiT was originally designed as an acceleration method for image synthesis. Here we have adapted it for video synthesis to facilitate comparison. Please refer to the Appendix for more details of the base models and compared methods.

**Evaluation metrics and datasets** To assess the performance of video synthesis acceleration methods, we focus primarily on two aspects, namely inference efficiency and visual quality. To evaluate inference efficiency, we employ Multiply-Accumulate Operations (MACs) and inference latency as metrics. We utilize VBench (Huang et al., 2024), LPIPS (Zhang et al., 2018), PSNR, and SSIM for visual quality evaluation. VBench is a comprehensive benchmark suit for video generative models. It is well-aligned with human perceptions and capable of providing valuable insights from multiple perspectives. LPIPS, PSNR, and SSIM measure the similarity between videos generated by the accelerated sampling method and those from the original model. PSNR quantifies pixel-level fidelity between outputs, LPIPS measures perceptual consistency, and SSIM assesses structural similarity. In general, higher similarity scores indicate better fidelity and visual quality.

**Implementation details** All experiments conduct full attention inference for spatial and temporal attention modules every 2 timesteps to facilitate dynamic feature reuse. The weight $w(t)$ increases linearly from 0 to 1 starting from the beginning of dynamic feature reuse until the end of sampling. For CFG output reuse, full inference is conducted every 5 timesteps, starting from $1/3$ of the total sampling steps (e.g., for Open-Sora 1.2, which has 30 total sampling steps, this begins at step 10). The hyperparameters $\alpha_1$ and $\alpha_2$ are set to a default value of 0.2, which performs well for most models. For more details on the selection of hyperparameters, please refer to Appendix A.5. All experiments are carried out on NVIDIA A100 80GB GPUs using PyTorch, with FlashAttention (Dao et al., 2022) enabled by default.

Table 1: Comparison of efficiency and visual quality on a single GPU.

| Method | Efficiency | | | Visual Quality | | | |
|---|---|---|---|---|---|---|---|
| | MACs (P) ↓ | Speedup ↑ | Latency (s) ↓ | VBench ↑ | LPIPS ↓ | SSIM ↑ | PSNR ↑ |
| **Open-Sora 1.2** (192 frames, 480P) | | | | | | | |
| Open-Sora 1.2 ($T=30$) | 6.30 | 1× | 192.07 | 78.79% | - | - | - |
| $\Delta$-DiT ($N_c=14, N=2$) | 5.51 | 1.14× | 168.69 | 77.43% | 0.2834 | 0.7403 | 17.77 |
| $\Delta$-DiT ($N_c=28, N=2$) | 4.72 | 1.34× | 143.14 | 76.60% | 0.3321 | 0.7092 | 16.24 |
| PAB | 5.33 | 1.23× | 156.73 | 78.15% | 0.1041 | 0.8821 | 26.43 |
| Ours | **4.13** | **1.62×** | **118.44** | **78.46%** | **0.0835** | **0.8932** | **27.03** |
| **Open-Sora-Plan** (65 frames, 512×512) | | | | | | | |
| Open-Sora-Plan ($T=150$) | 10.30 | 1× | 103.76 | 80.16% | - | - | - |
| $\Delta$-DiT ($N_c=14, N=3$) | 8.60 | 1.19× | 86.88 | 78.12% | 0.4515 | 0.4813 | 16.08 |
| $\Delta$-DiT ($N_c=28, N=3$) | 6.90 | 1.46× | 70.99 | 77.71% | 0.4819 | 0.4467 | 15.42 |
| PAB | 7.39 | 1.32× | 78.72 | 80.06% | 0.2423 | 0.7126 | 20.29 |
| Ours | **5.51** | **1.68×** | **61.68** | **80.19%** | **0.1348** | **0.8138** | **23.72** |
| **Latte** (16 frames, 512×512) | | | | | | | |
| Latte ($T=50$) | 3.05 | 1× | 29.22 | 77.05% | - | - | - |
| $\Delta$-DiT ($N_c=14, N=2$) | 2.67 | 1.23× | 23.80 | 76.27% | 0.1731 | 0.8107 | 22.69 |
| $\Delta$-DiT ($N_c=28, N=2$) | 2.29 | 1.43× | 20.38 | 76.01% | 0.2245 | 0.7620 | 21.00 |
| PAB | 2.24 | 1.28× | 22.84 | 76.70% | 0.2904 | 0.7083 | 18.98 |
| Ours | **1.97** | **1.54×** | **18.98** | **76.89%** | **0.0817** | **0.8948** | **28.21** |
| **CogVideoX** (48 frames, 480P) | | | | | | | |
| CogVideoX ($T=50$) | 6.03 | 1× | 78.48 | 80.18% | - | - | - |
| $\Delta$-DiT ($N_c=4, N=2$) | 5.62 | 1.08× | 72.72 | 79.61% | 0.3319 | 0.6612 | 17.93 |
| $\Delta$-DiT ($N_c=8, N=2$) | 5.23 | 1.15× | 68.19 | 79.31% | 0.3822 | 0.6277 | 16.69 |
| $\Delta$-DiT ($N_c=12, N=2$) | 4.82 | 1.26× | 62.50 | 79.09% | 0.4053 | 0.6126 | 16.15 |
| PAB | 4.45 | 1.35× | 57.98 | 79.76% | 0.0860 | 0.8978 | 28.04 |
| Ours | **3.71** | **1.62×** | **48.44** | **79.83%** | **0.0766** | **0.9066** | **28.93** |
| **Vchitect-2.0** (40 frames, 480P) | | | | | | | |
| Vchitect-2.0 ($T=100$) | 14.57 | 1× | 260.32 | 80.80% | - | - | - |
| $\Delta$-DiT ($N_c=6, N=3$) | 13.00 | 1.11× | 233.59 | 79.98% | 0.4153 | 0.5837 | 14.26 |
| $\Delta$-DiT ($N_c=12, N=3$) | 11.79 | 1.24× | 209.78 | 79.50% | 0.4534 | 0.5519 | 13.68 |
| PAB | 12.20 | 1.26× | 206.23 | 79.56% | 0.0489 | 0.8806 | 27.38 |
| Ours | **8.67** | **1.67×** | **156.13** | **80.84%** | **0.0282** | **0.9224** | **31.45** |

## 3.2 MAIN RESULTS

**Quantitative comparison**   Table 1 presents a quantitative comparison of our method with $\Delta$-DiT and PAB in terms of efficiency and visual quality. We synthesize videos with prompts provided by VBench and use the synthesized videos to compute the VBench metrics as well as calculate LPIPS, SSIM, and PSNR with videos sampled by the original model. The results demonstrate that our method achieves stable acceleration efficiency and superior visual quality across different base models, sampling schedulers, video resolutions, and lengths.

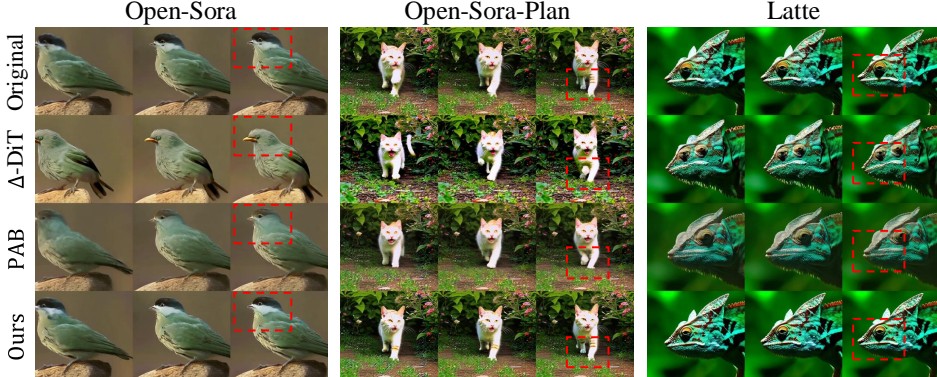

Figure 9: Visual quality comparison of different methods. Differences are highlighted in red boxes.

**Visual quality comparison**   Fig. 9 compares the videos generated by our method against those by the original model, PAB, and $\Delta$-DiT. The results demonstrate that our method can effectively preserve the original quality and fine details. More visual results can be found in the Appendix.

## 3.3 ABLATION STUDY

To comprehensively assess the effectiveness and efficiency of our method, we perform extensive ablation studies based on Open-Sora, synthesizing videos of 48 frames at 480P.

**Efficiency**   Table 2 compares the efficiency of the original Open-Sora and its variants with different acceleration components. There are two key observations. (1) The Dynamic Feature Reuse Strategy and CFG-Cache independently contribute to significant reductions in inference costs. When combined, they further minimize inference overhead. (2) Compared to Vanilla Feature Reuse, the proposed Dynamic Feature Reuse strategy has a negligible impact on efficiency.

Table 2: Impact on inference efficiency.

| Variants | | | MACs (P) | Latency (s) | $\Delta$ (s) |
|---|---|---|---|---|---|
| Vanilla FR | Dynamic FR | CFG-Cache | | | |
| | | | 1.54 | 41.28 | - |
| ✓ | | | 1.33 | 33.25 | -8.03 |
| | ✓ | | 1.33 | 33.50 | -7.78 |
| | | ✓ | 1.16 | 31.32 | -9.96 |
| | ✓ | ✓ | 1.01 | 26.12 | -15.16 |

(Vanilla FR denotes Vanilla Feature Reuse, and $\Delta$ represents the reduction in latency compared to the original model.)

**Visual quality**   Table 3 compares the visual quality of the original Open-Sora with its variants implementing different acceleration components. Note that vanilla feature reuse leads to a performance drop in VBench and LPIPS. The introduction of the dynamic feature reuse strategy mitigates the loss of information and thereby improves the performance of these metrics (*e.g.*, VBench: 78.34% $\rightarrow$ 78.69%). Fig. 10 (a) provides a visual comparison of the results. It can be observed that vanilla feature reuse shows reduced details (*e.g.*, the moon and snowflakes), whereas dynamic feature reuse strategy can significantly alleviate this problem. The Feature MSE curves show that adding the bias term can lower the MSE between intermediate features from the original and accelerated sampling process, aligning with the visual results.

Table 3: Impact on visual quality.

| Variants | VBench | LPIPS | PSNR | SSIM |
|---|---|---|---|---|
| Original Open-Sora | 78.99% | - | - | - |
| Full (w/ Vanilla FR) | 78.34% | 0.0657 | 28.20 | 0.8785 |
| Full (w/ Dynamic FR) | 78.69% | 0.0590 | 28.41 | 0.8938 |
| CFG-Cache w/o Enhancement | 78.42% | 0.0709 | 27.97 | 0.8727 |
| Enhance LF only | 78.58% | 0.0617 | 28.29 | 0.8894 |
| Enhance HF only | 78.49% | 0.0686 | 28.08 | 0.8834 |
| Full (w/ full CFG-Cache) | 78.69% | 0.0590 | 28.41 | 0.8938 |

(FR denotes Feature Reuse.)

Table 4: Scaling to multiple GPUs with DSP.

| Method | 1× A100 | 2× A100 | 4× A100 | 8× A100 |
|---|---|---|---|---|
| **Open-Sora** ( 192 frames, 480P) | | | | |
| Open-Sora | 192.07 (1×) | 72.82 (2.64×) | 39.09 (4.92×) | 21.62(8.89×) |
| PAB | 156.73 (1.23×) | 58.11(3.31×) | 30.91 (6.21×) | 17.21 (11.16×) |
| Ours | 118.44 (1.62×) | 42.18(4.55×) | 22.55 (8.52×) | 12.57 (15.28×) |
| **Open-Sora-Plan**(221 frames, 512×512) | | | | |
| Open-Sora-Plan | 316.71 (1×) | 169.21 (1.87×) | 89.10 (3.55×) | 49.13(6.44×) |
| PAB | 243.33 (1.30×) | 127.30 (2.49×) | 71.17 (4.45×) | 37.13(8.53×) |
| Ours | 187.91 (1.69×) | 104.37 (3.03×) | 57.70 (5.49×) | 31.82(9.95×) |

Referring to Table 3, it can be seen that introducing CFG-Cache without enhancement reduces the visual quality. On the other hand, CFG-Cache with dynamic enhancement of either the low- or high-

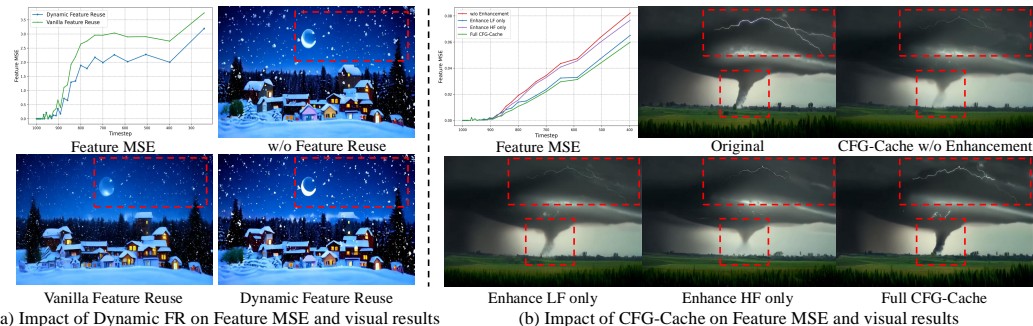

(a) Impact of Dynamic FR on Feature MSE and visual results | (b) Impact of CFG-Cache on Feature MSE and visual results

Figure 10: Comparison of Feature MSE curves and visual results from the ablation study.

frequency bias helps to improve the visual quality, and their combined effect achieves the best visual quality. Fig. 10 (b) shows that enhancing low-frequency bias improves the fidelity of low-frequency components (e.g., clouds, tornado outlines) while enhancing high-frequency bias enriches high-frequency details (e.g., lightning). The Feature MSE curve of CFG-Cache without enhancement aligns with the reduced visual quality. Dynamic enhancement helps to mitigate error accumulation, leading to higher visual fidelity.

### 3.4 SCALABILITY AND GENERALIZATION

**Scaling to multiple GPUs**  To evaluate the sampling efficiency of our method on multiple GPUs, we adopt the approach used in PAB and integrate Dynamic Sequence Parallelism (DSP) (Zhao et al., 2024b) to distribute the workload across GPUs. Table 4 illustrates that, as the number of GPUs increases, our method consistently enhances inference speed across different base models, surpassing the performance of the compared methods.

**Performance at different resolutions and lengths**  To evaluate the effectiveness of our method in accelerating sampling for videos of varying sizes, we conduct tests across different video lengths and resolutions and report the results in Fig. 11. Our method maintains stable acceleration performance when faced with increasing resolutions and frame counts in videos, demonstrating its potential to accelerate sampling longer and higher-resolution videos in line with practical demands.

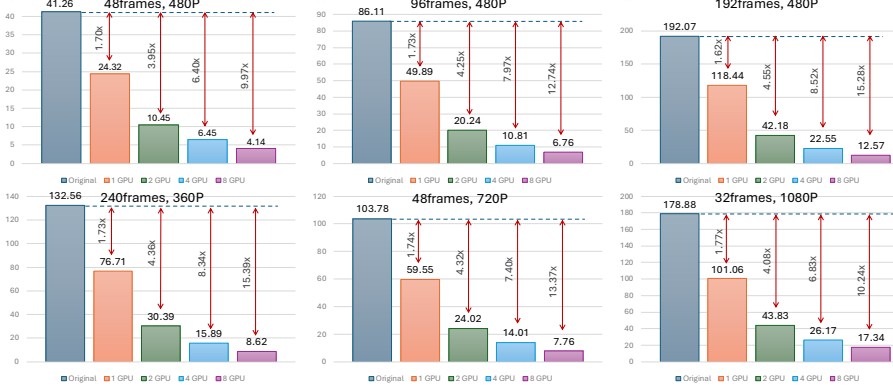

Figure 11: Acceleration efficiency of our method at different video resolutions and lengths.

**I2V and image synthesis performance**  We integrate our acceleration method to the state-of-the-art image-to-video model DynamiCrafter (Xing et al., 2023) and image synthesis model PixArt-sigma (Chen et al., 2024a). As shown in Fig. 12, our method significantly accelerates sampling while maintaining visual fidelity, demonstrating its potential for extension to various base models.

## 4 RELATED WORK

### 4.1 DIFFUSION MODELS FOR VIDEO SYNTHESIS

Diffusion models have demonstrated potential in high-quality image synthesis (Ho et al., 2020; Rombach et al., 2022; Chen et al., 2023; 2024b), attracting significant attention. Subsequent works have adapted these models for video synthesis to generate high-fidelity videos (Ho et al., 2022). Motivated by advancements in image synthesis, early studies typically employed the diffusion UNet architecture (Blattmann et al., 2023; Wang et al., 2023; Zhang et al., 2023; Wu et al., 2023; Zhang et al., 2024c). As the scalability of diffusion transformer (Peebles & Xie, 2023) was validated in image synthesis, an increasing number of works have adopted the diffusion transformer as the noise

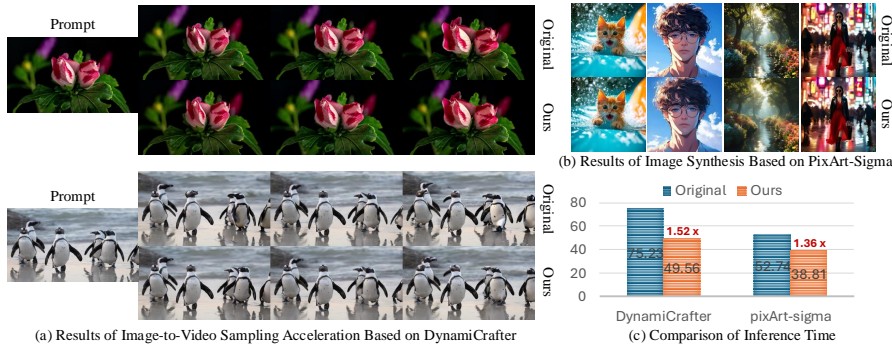

Figure 12: Visual results and inference time of our method on I2V and image synthesis models.

estimation network (Ma et al., 2024a; Zheng et al., 2024; PKU-Yuan Lab and Tuzhan AI etc., 2024; Yang et al., 2024).

### 4.2 EFFICIENCY IMPROVEMENTS IN DIFFUSION MODELS

Despite the impressive performance of diffusion models in image and video synthesis, their substantial inference cost limits their practicality. Prior research on improving the efficiency of diffusion models has primarily focused on two perspectives, namely reducing the number of sampling steps and lowering the inference cost per sampling step. Regarding the reduction of sampling steps, most approaches achieve high-quality samples with fewer steps by employing efficient SDE or ODE solvers (Song et al., 2020; Lu et al., 2022a;b). Other methods reduce sampling steps by progressively distilling the model (Salimans & Ho, 2022; Meng et al., 2023; Sauer et al., 2023; Lin & Yang, 2024; Li et al., 2024b) or employing consistency models (Luo et al., 2023; Song et al., 2023).

More works have focused on reducing the inference cost per timestep. Some approaches improve network efficiency through pruning (Zhang et al., 2024a) or quantization (Shang et al., 2023; So et al., 2024a; He et al., 2024; Li et al., 2024a; Sui et al., 2024; Zhao et al., 2024a), while others obtain more lightweight network architectures through search techniques (Li et al., 2023a; Yang et al., 2023). However, these methods often require additional computational resources for fine-tuning or optimization. Some training-free approaches (Bolya & Hoffman, 2023; Wang et al., 2024) focus on the input tokens, accelerating the sampling process by reducing the number of tokens to be processed by eliminating token redundancy in image synthesis. Other methods reuse intermediate features between adjacent sampling timesteps, avoiding redundant computations (Wimbauer et al., 2024; So et al., 2024b). TGATE (Zhang et al., 2024b) accelerates image generation by caching and reusing attention outputs at scheduled timesteps. DeepCache (Ma et al., 2024b) and Faster Diffusion (Li et al., 2023b) employ a feature caching mechanism to indirectly alter the UNet diffusion for acceleration. $\Delta$-DiT (Chen et al., 2024c) adapts this mechanism to the diffusion transformer architecture by caching the residuals between attention layers. PAB (Zhao et al., 2024c) caches and broadcasts intermediate features at different timestep intervals based on the characteristics of varying attention blocks. Although these methods have achieved some improvements in diffusion efficiency, the efficiency enhancements for diffusion transformers in video synthesis remain insufficient.

## 5 CONCLUSION AND DISCUSSION

In this work, we present *FasterCache*, a training-free strategy that significantly accelerates video synthesis inference while preserving high-quality generation. Through analysis of existing cache-based methods, we find that directly reusing adjacent-step features in attention modules can degrade video quality. Additionally, we investigate the acceleration potential of CFG, identifying redundancy between conditional and unconditional features at the same timestep. Leveraging these insights, *FasterCache* integrates a dynamic feature reuse strategy that maintains feature distinction and temporal continuity, and CFG-Cache which optimizes the reuse of conditional and unconditional outputs to further boost speed without sacrificing detail quality. Extensive experiments demonstrate its strong performance in both efficiency and synthesis quality across diverse video models, sampling schedules, video lengths and resolutions, highlighting its potential for real-world applications.

**Limitation** Despite the effectiveness shown by our method, certain limitations remain. When the synthesis quality of the model is suboptimal, our acceleration method is unlikely to yield satisfactory results either. We believe that advancements in base video models will mitigate this issue. Additionally, in complex scenes with substantial video motion, our method may occasionally produce degraded results. At present, this can be remedied through manual adjustments of hyperparameters. In the future, we plan to investigate strategies for adaptive caching to further enhance performance.

## 6 ACKNOWLEDGEMENTS

This study is supported by the National Key R&D Program of China No.2022ZD0160102, and by the video generation project (Intern-Vchitect) of Shanghai Artificial Intelligence Laboratory. This study is also supported by the Ministry of Education, Singapore, under its MOE AcRF Tier 2 (MOET2EP20221-0012, MOE-T2EP20223-0002), and under the RIE2020 Industry Alignment Fund – Industry Collaboration Projects (IAF-ICP) Funding Initiative, as well as cash and in-kind contribution from the industry partner(s).

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

## A APPENDIX

### A.1 FURTHER DETAILS OF BASE MODELS

In this work, we applied our FasterCache to various video synthesis models, including Open-Sora 1.2 (Zheng et al., 2024), Open-Sora-Plan (PKU-Yuan Lab and Tuzhan AI etc., 2024), Latte (Ma et al., 2024a), CogVideoX (Yang et al., 2024), and Vchitect 2.0 (Fan et al., 2025). Open-Sora 1.2 integrates 2D-VAE and 3D-VAE to enhance video compression and employs ST-DiT blocks for the diffusion process. Open-Sora-Plan adopts CausalVideoVAE to compress visual representations better and 3D full attention architecture to capture joint spatial and temporal features. Latte extracts spatio-temporal tokens from input videos and then adopts a series of transformer blocks to model video distribution in the latent space. CogVideoX employs a 3D VAE to compress videos along spatial and temporal dimensions and an expert transformer with the expert adaptive LayerNorm to facilitate the fusion between the two modalities.

### A.2 FURTHER DETAILS OF COMPARED METHODS

**PAB** (Zhao et al., 2024c) employs a pyramid-style broadcasting mechanism to propagate attention outputs across subsequent steps. It optimizes efficiency by applying distinct broadcast strategies to each attention layer based on their respective variances. Additionally, the method introduces broadcast sequence parallelism to enhance the efficiency of distributed inference. This paper follows the default parameter configuration of PAB.

**$\Delta$-DiT** (Chen et al., 2024c) accelerates inference by caching feature offsets instead of the full feature maps while preventing input information loss. It caches the residuals of the blocks in the latter part of DiT for approximation during early-stage sampling and caches the residuals of the blocks in the earlier part during later-stage sampling. In $\Delta$-DiT, the parameters that need to be configured are the residual cache interval $N$, the number of cached blocks $N_c$, and the timestep boundary $b$ for determining the position of the cached blocks. Since the source code of $\Delta$-DiT is not publicly available, we implemented its method based on the paper for accelerating video synthesis. Following the guidelines in $\Delta$-DiT, we experimented with different configurations of $N_c$ and $N$ to balance visual quality and inference speed, allowing for a fair evaluation of the method.

### A.3 MORE DISCUSSION

#### A.3.1 MORE DISCUSSION ON DYNAMIC FEATURE REUSE

**Effectiveness of Dynamic Feature Reuse**    Assume that the output features of a particular layer in the diffusion model are a function of the timestep $t$, denoted as $F(t)$. The motivation behind Vanilla Feature Reuse lies in the observation that features at adjacent timesteps are highly similar. Vanilla Feature Reuse avoids the computation at the current timestep by directly reusing the features from the previous timestep, i.e. $F(t) = F(t + \Delta t)$. Although $F(t)$ and $F(t + \Delta t)$ are very close with a minimal error $E = F(t) - F(t + \Delta t)$, the difference is not zero. To estimate this error, we assume that $F(t)$ is a smooth and differentiable function with respect to $t$, allowing us to perform a Taylor expansion, yielding:

$$F(t + \Delta t) = F(t) + \frac{dF(t)}{dt}\Delta t + \frac{d^2 F(t)}{dt^2}\frac{\Delta t^2}{2} + O(\Delta t^3), \tag{12}$$

$$F(t + 3\Delta t) = F(t) + 3\frac{dF(t)}{dt}\Delta t + 3\frac{d^2 F(t)}{dt^2}\frac{\Delta t^2}{2} + O(\Delta t^3). \tag{13}$$

By subtracting these expansions, we derive:

$$F(t + \Delta t) - F(t + 3\Delta t) = (\frac{dF(t)}{dt}\Delta t) \times (-2) + O(\Delta t^2), \tag{14}$$

Based on the statistics of approximately 200 video samples, we plotted the magnitudes of the first-order and second-order terms of $F(t)$. When $\Delta t$ (e.g., $\Delta t = 1$) is sufficiently small, the norm of second-order term is smaller than that of the first-order term, as shown in Fig. 13 (c). Furthermore, we tested three different estimations for $F(t)$, denoted as $\hat{F}(t)$: (a) $\hat{F}(t) = F(t + 1)$, (b) $\hat{F}(t) = F(t + 1) - \frac{dF(t)}{dt}$, and (c) $\hat{F}(t) = F(t + 1) - \frac{dF(t)}{dt} - \frac{d^2 F(t)}{2dt^2}$. Subsequently, we calculated the L1

distance between each $\hat{F}(t)$ and $F(t)$. As shown in the Fig. 13 (d), incorporating the second-order term yields only a marginal reduction in the L1 distance compared to the first-order term. Therefore, the second-order terms contribute only marginally to the improvement in visual quality (VBench: 78.77% $\rightarrow$ 78.80%). However, the computation of second-order terms incurs significant costs in memory and latency. **Considering both simplicity and efficiency**, we use only the first-order term for error estimation in Dynamic Feature Reuse. Based on these analyses and statistical results, we define the error term as:

$$E = F(t) - F(t + \Delta t) \approx -\frac{dF(t)}{dt}\Delta t = (F(t + \Delta t) - F(t + 3\Delta t)) * w. \quad (15)$$

The scale factor $w$ is introduced to scale the bias term to approximate the error $E$. In Eq. (5), $E = F_{t-1} - F_{cache}^t \approx (F_{cache}^t - F_{cache}^{t+2}) * w(t)$. By introducing this feature bias term, the information loss could be reduced, thereby improving the quality of the synthesis videos while maintaining computational efficiency.

**Design choices for $w(t)$ in Dynamic Feature Reuse** As shown in Fig. 13 (a), we tried different design choices for Dynamic Feature Reuse (DFR) and found that the linear increasing strategy is a simple and effective manner for dynamically capturing missing features. Different design choices for DFR: (1) Constant weights $w(t)$. A constant weight of $w(t) = 0.5$ is applied to the feature biases at each accelerated timesteps. (2) Learnable weights $w(t)$. We introduced a set of learnable parameters $w(t)$, which are optimized by minimizing the MSE loss between the features output by DFR during accelerated sampling and those generated in the original unaccelerated sampling process, resulting in the learned $w(t)$. (3) Linearly increasing $w(t)$ (Our DFR). Starting from the application of DFR to the end of sampling proces, the weight function $w(t)$, used for weighting feature biases, linearly increases from 0 to 1.

The trend of the optimized $w(t)$ is shown in Fig 13 (a), the result indicates that $w(t)$ obtained through optimization gradually increases as sampling progresses. This trend is primarily attributed to the increasing stability of feature biases in Eq. 5 with respect to the sampling timesteps and the growing reliance on bias features for synthesizing high-quality details in the later stages of sampling. The performance of different strategies is shown in Table 5. All results incorporating feature biases outperform those without them. The linearly increasing $w(t)$ achieves comparable performance to optimized learnable $w(t)$, both outperforming constant $w(t)$. Given the simplicity of linear interpolation, we ultimately adopt linearly interpolated $w(t)$ to weight the feature biases.

Table 5: Performance of different Dynamic FR strategies.

| Variants | LPIPS | PSNR | SSIM |
|---|---|---|---|
| Vanilla FR | 0.0657 | 28.20 | 0.8785 |
| Dynamic FR (Constant $w(t) = 0.5$) | 0.0615 | 28.33 | 0.8889 |
| Dynamic FR(Learned $w(t)$) | 0.0596 | **28.45** | **0.8941** |
| Dynamic FR(Linear $w(t)$) | **0.0590** | 28.41 | 0.8938 |

**Comparison between Dynamic FR and Vanilla FR** Fig. 13 (b) presents the generated results of Vanilla Feature Reuse (FR) and Dynamic FR and the differences between the features produced by Vanilla FR and Dynamic FR compared to the original features. It is evident that, due to the introduction of feature biases, the feature differences between Dynamic FR and the original features are less significant. In contrast, the features produced by the model accelerated with Vanilla FR exhibit detail loss compared to the original features, leading to noticeable detail degradation in the synthesized images (as highlighted by the red box).

### A.3.2 FURTHER DISCUSSION ON CFG-CACHE

**Effectiveness of CFG-Cache** The reliability of CFG-Cache stems from three key factors: (a) After the early stage $t_{early}$, the similarity between conditional output $cond(t)$ and unconditional output $uncond(t)$ at the same timestep $t$:

$$uncond(t) = cond(t) + \Delta, when\ t >= t_{early}. \quad (16)$$

(b) The predictability of biases between conditional and unconditional output from previous timesteps, expressed as:

$$\Delta = uncond(t + \Delta t) - cond(t + \Delta t) = uncond(t) - cond(t) + \epsilon. \quad (17)$$

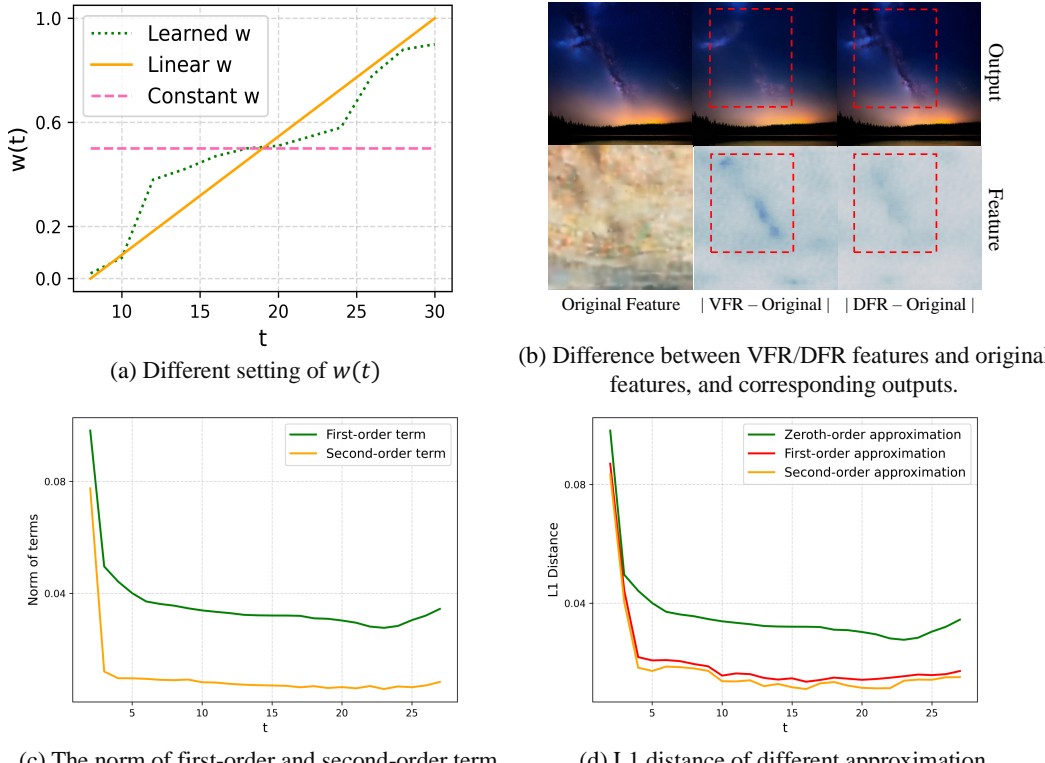

(a) Different setting of $w(t)$

(b) Difference between VFR/DFR features and original features, and corresponding outputs.

(c) The norm of first-order and second-order term

(d) L1 distance of different approximation.

Figure 13: Design choices for Dynamic Feature Reuse and comparison between Dynamic Feature Reuse (DFR) and Vanilla Feature Ruse (VFR).

In practice, we find that when $\Delta t$ is sufficiently small, the $\epsilon$ can be considered negligible. Then:

$$uncond(t) \approx cond(t) + (uncond(t + \Delta t) - cond(t + \Delta t)) \qquad (18)$$

(c) The dynamic variations of the frequency-domain distribution of feature biases, as illustrated in Fig. 7(b) and Fig.14.

**Visualization of CFG biases**  From the onset of CFG-Cache to the end of sampling, the differences between the conditional and unconditional output features progressively shift from being dominated by low-frequency features to high-frequency features. As shown in Fig 14, this observation aligns with the feature visualization analysis: during the early and middle sampling stages, CFG primarily guides the model to synthesize perceptual features such as reasonable shapes and layouts, which are often represented in the low-frequency feature domain. In contrast, during the later stages of sampling, CFG contributes primarily to the synthesis of high-quality details, typically governed by high-frequency features. This insight motivates us to assign higher weights to features of different frequencies at different stages, allowing to gain more emphasis, thereby preserving the visual quality.

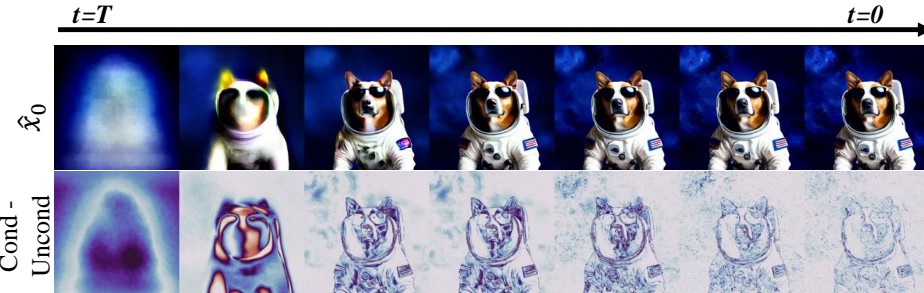

Figure 14: The variation in differences between the conditional and unconditional outputs during the sampling process.

### A.3.3 FASTERCACHE UNDER DIFFERENT CFG SCALES AND NEGATIVE PROMPTS

We compared two different negative prompt settings on Open-Sora: (1) default empty negative prompt and (2) non-empty negative prompt:

*"worst quality, normal quality, low quality, low res, blurry, text, watermark, logo, banner, extra digits, cropped, jpeg artifacts, signature, username, error, sketch, duplicate, ugly, monochrome, horror, geometry, mutation, disgusting, bad anatomy, bad proportions, bad quality, deformed, disconnected limbs, out of frame, out of focus, dehydrated, disfigured, extra arms, extra limbs, extra hands, fused fingers, gross proportions, long neck, jpeg, malformed limbs, mutated, mutated hands, mutated limbs, missing arms, missing fingers, picture frame, poorly drawn hands, poorly drawn face, collage, pixel, pixelated, grainy"*

We calculated the LPIPS, SSIM, and PSNR between the videos generated by FasterCache and those generated by the original model. As shown in Fig. 15 (a) and (b), the experimental results show that FasterCache performs similarly under both prompt settings. This is consistent with our expectations, as CFG-Cache caches the biases between the conditional and unconditional outputs, which are not significantly affected by changes in the negative prompt setting.

We also experimented with different CFG guidance scales $g$ on Open-Sora. As shown in Fig 15 (b) and (c), regardless of increasing or decreasing the scale, while the adjustment affects the original Open-Sora results, FasterCache consistently maintains a high level of alignment with the original results, particularly in preserving details. Therefore, FasterCache is not affected by changes in the CFG guidance scale and maintains high-quality acceleration.

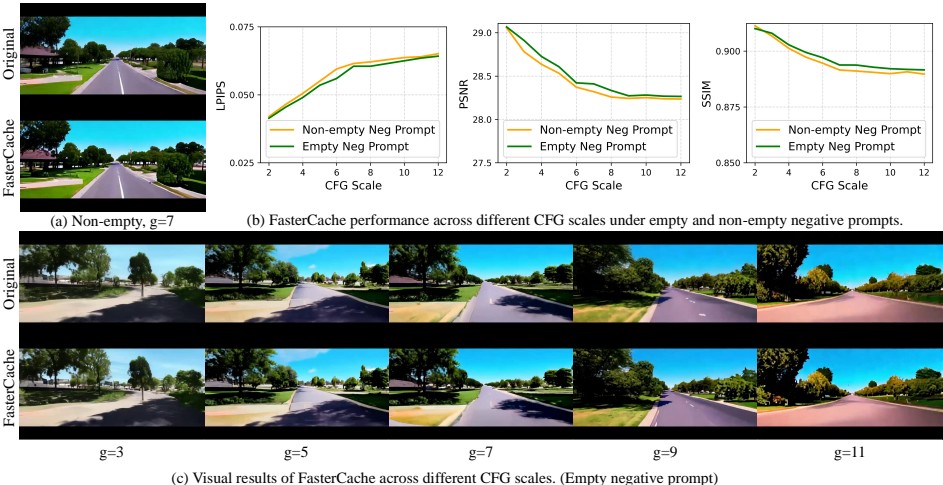

Figure 15: The performance of FasterCache under different CFG scales with empty and non-empty negative prompt settings.

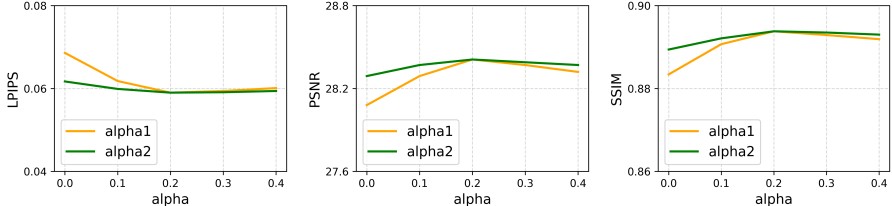

Figure 16: Different Settings of $\alpha$ in CFG-Cache.

### A.4 ADDITIONAL QUALITATIVE EXPERIMENTS

**More visual results on Text-to-Video models** The additional visual comparison results for Open-Sora 1.2 (Zheng et al., 2024), Open-Sora-Plan (PKU-Yuan Lab and Tuzhan AI etc., 2024), and Latte (Ma et al., 2024a) are presented in Fig. 17, Fig. 18, and Fig. 19, while further comparisons for CogVideoX-2B (Yang et al., 2024) and Vchitect-2.0 (Fan et al., 2025) are shown in Fig. 20. Our method demonstrates reliable fidelity across various models and styles or content in video synthesis, while simultaneously achieving acceleration.

Additionally, Fig. 21 demonstrates the visual performance of FasterCache on state-of-the-art models CogVideoX-5B and Mochi-10B (Team, 2024). FasterCache achieves an acceleration of 1.63 times (206s → 126s) on CogVideoX-5B and 1.74 times (320s → 184s) on Mochi-10B. As model scale increases, FasterCache consistently accelerates the sampling process while maintaining fidelity in synthesized videos. We also observe that as the generative capability of the base model improves, FasterCache becomes more robust in synthesizing videos with complex scenes or rapid motion. For instance, in Fig. 21, the $1st$ example shows subtle details of small groups of fish, the $3rd$ example highlights intricate finger details and complex non-rigid motions, and the $4th$ and $5th$ examples exhibit rapid and large-scale movements. These results demonstrate the broad potential of FasterCache in practical applications.

**More visual results on Image-to-Video models**  We conducted image-to-video sampling acceleration experiments based on DynamiCrafter (Xing et al., 2023), achieving a $1.52\times$ speedup on a single GPU. Additional visual results are provided in Fig. 22. Our method demonstrates good fidelity in the acceleration of image-to-video models, indicating broad potential for practical applications.

## A.5  ADDITIONAL QUANTITATIVE EXPERIMENTS

### A.5.1  USER PREFERENCE STUDY

To assess the effectiveness of our FasterCache, we additionally conduct a human evaluation. We randomly selected 30 videos for each model. Each rater receives a text prompt and two generated videos from different sampling acceleration methods (in random order). They are then asked to select the video with better visual quality. Five raters evaluate each sample, and the voting results are summarized in Table 6. As one can see, compared to other acceleration methods, the raters strongly prefer the videos generated by our method.

Table 6: User preference study. The numbers represent the percentage of raters who favor the videos synthesized by our method.

| Method comparison | Open-Sora 1.2 | Open-Sora-Plan | Latte |
|---|---|---|---|
| Ours vs. $\Delta$-DiT | 80.67% | 78.00% | 77.33% |
| Ours vs. PAB | 69.33% | 72.67% | 74.00% |

### A.5.2  HYPERPARAMETER SELECTION

Table 7: Different Dynamic FR caching intervals. Table 8: Different CFG-Cache caching intervals.

| Interval | LPIPS | PSNR | SSIM |
|---|---|---|---|
| 2 | 0.0590 | 28.41 | 0.8938 |
| 3 | 0.0698 | 27.95 | 0.8853 |
| 4 | 0.0751 | 27.61 | 0.8823 |
| 5 | 0.0897 | 27.39 | 0.8712 |

| Interval | LPIPS | PSNR | SSIM |
|---|---|---|---|
| 1 | 0.0496 | 28.88 | 0.8964 |
| 3 | 0.0537 | 28.56 | 0.8947 |
| 5 | 0.0590 | 28.41 | 0.8938 |
| 7 | 0.0724 | 27.68 | 0.8818 |
| 9 | 0.0104 | 27.44 | 0.8706 |

**Caching timestep interval of Dynamic Feature Reuse**  We experimented with different caching timestep intervals for Dynamic Feature Reuse. According to Table 7, it can be observed that as the caching timestep interval increases, the fidelity of the synthesized results gradually decreases. In practice, the caching timestep interval for Dynamic Feature Reuse can be adjusted as needed.

**Caching timestep interval of CFG-Cache** We experimented with different CFG-Cache intervals and found that when the interval exceeds 5 timesteps, there is a significant decline in fidelity, as shown in Table 8. Therefore, to balance fidelity and efficiency, we chose a CFG-Cache caching interval of 5. This means that after CFG-Cache is initiated, the model performs full inference for both the conditional and unconditional branches every 5 timesteps and caches the features.

**The configuration of $\alpha$ in CFG-Cache.**  In CFG-Cache, we experimented with different configurations of $\alpha$, where $\alpha_1$ is used to enhance low-frequency biases and $\alpha_2$ is used to enhance high-frequency biases. Through these experiments shown in Fig. 16, we found that $\alpha_1 = 0.2$ and $\alpha_2 = 0.2$ works effectively.

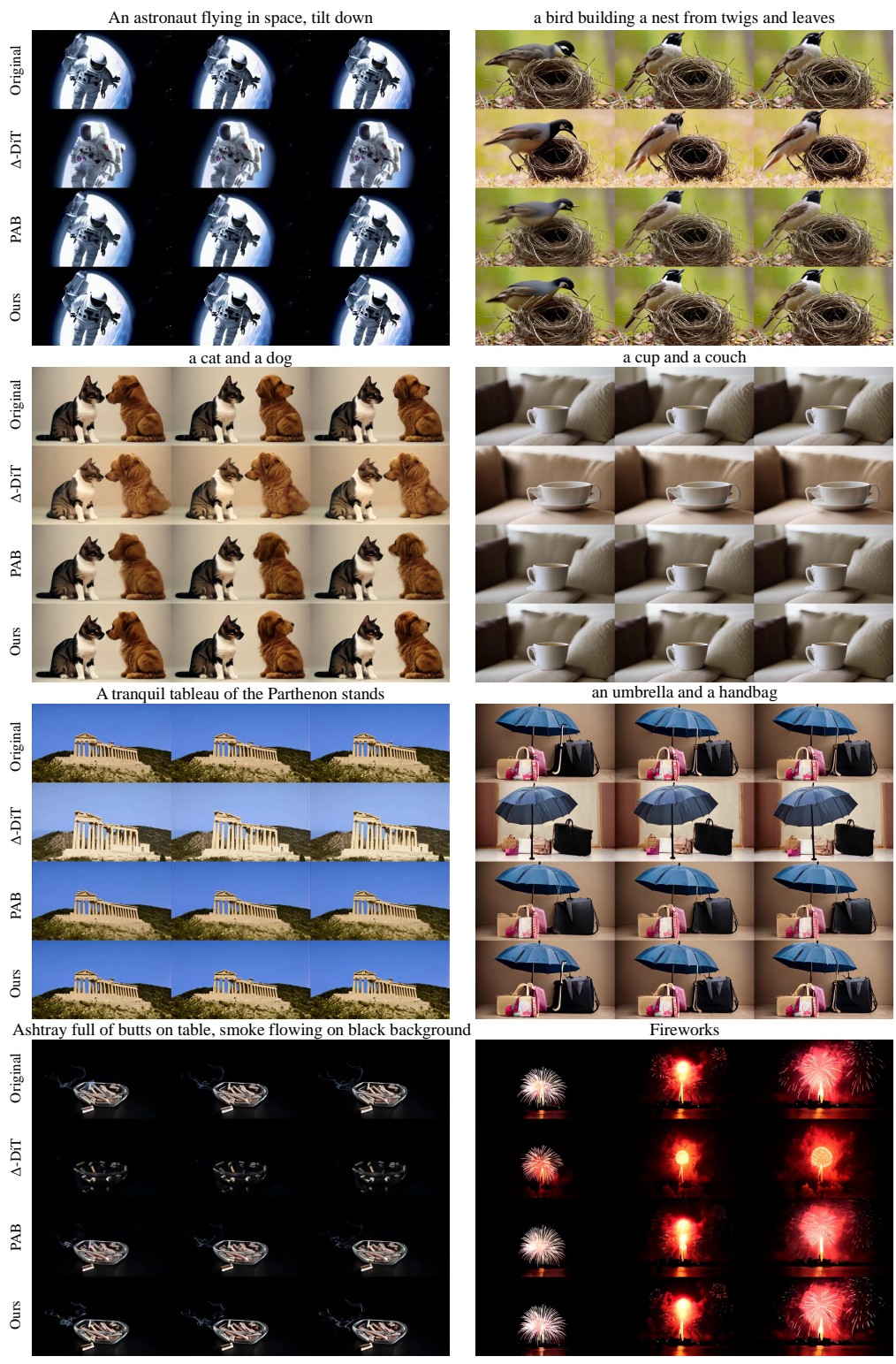

Figure 17: More visual results on Open-Sora (480P 192 frames). Zoom in for details.

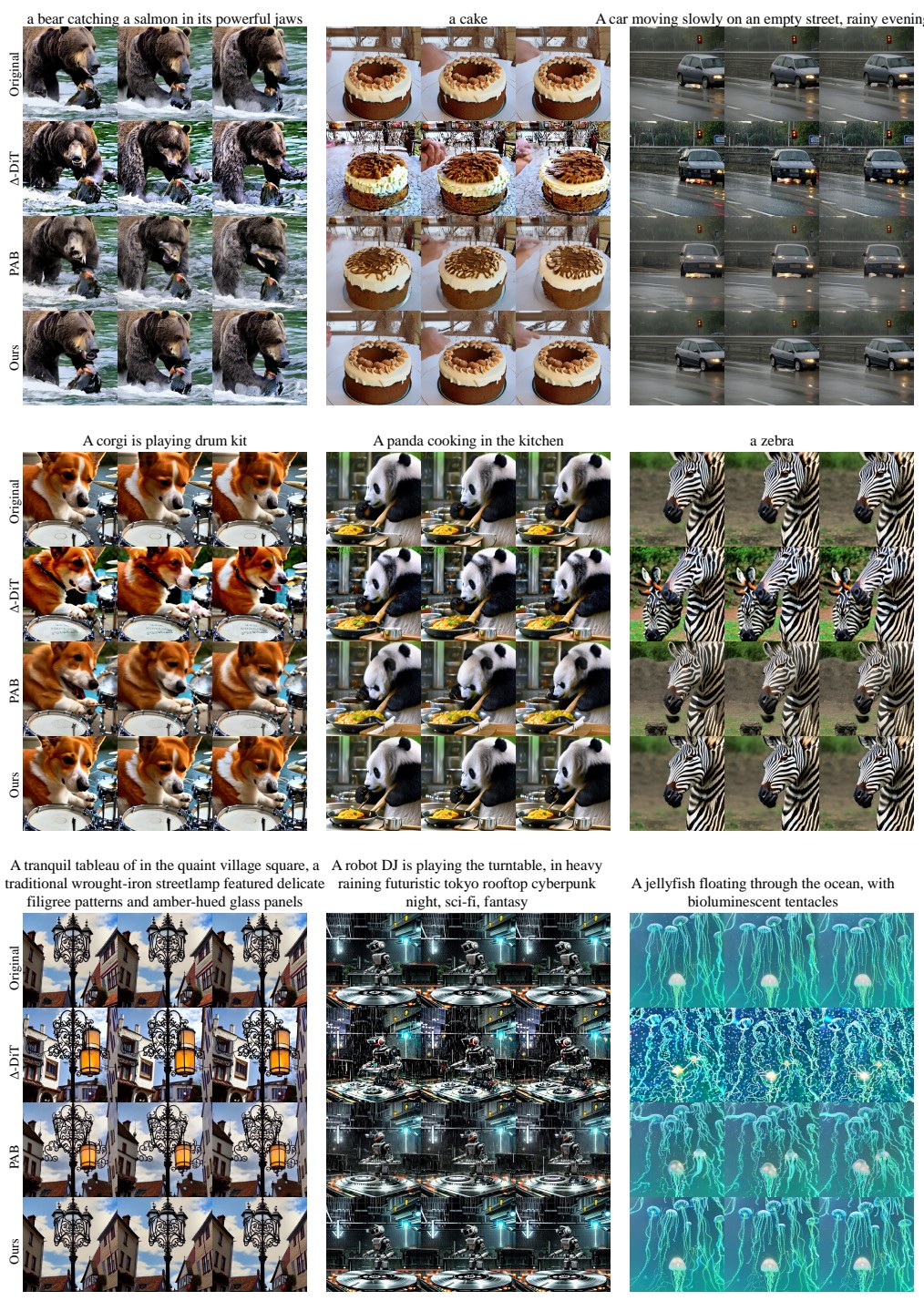

Figure 18: More visual results on Open-Sora-Plan (512×512 65 frames). Zoom in for details.

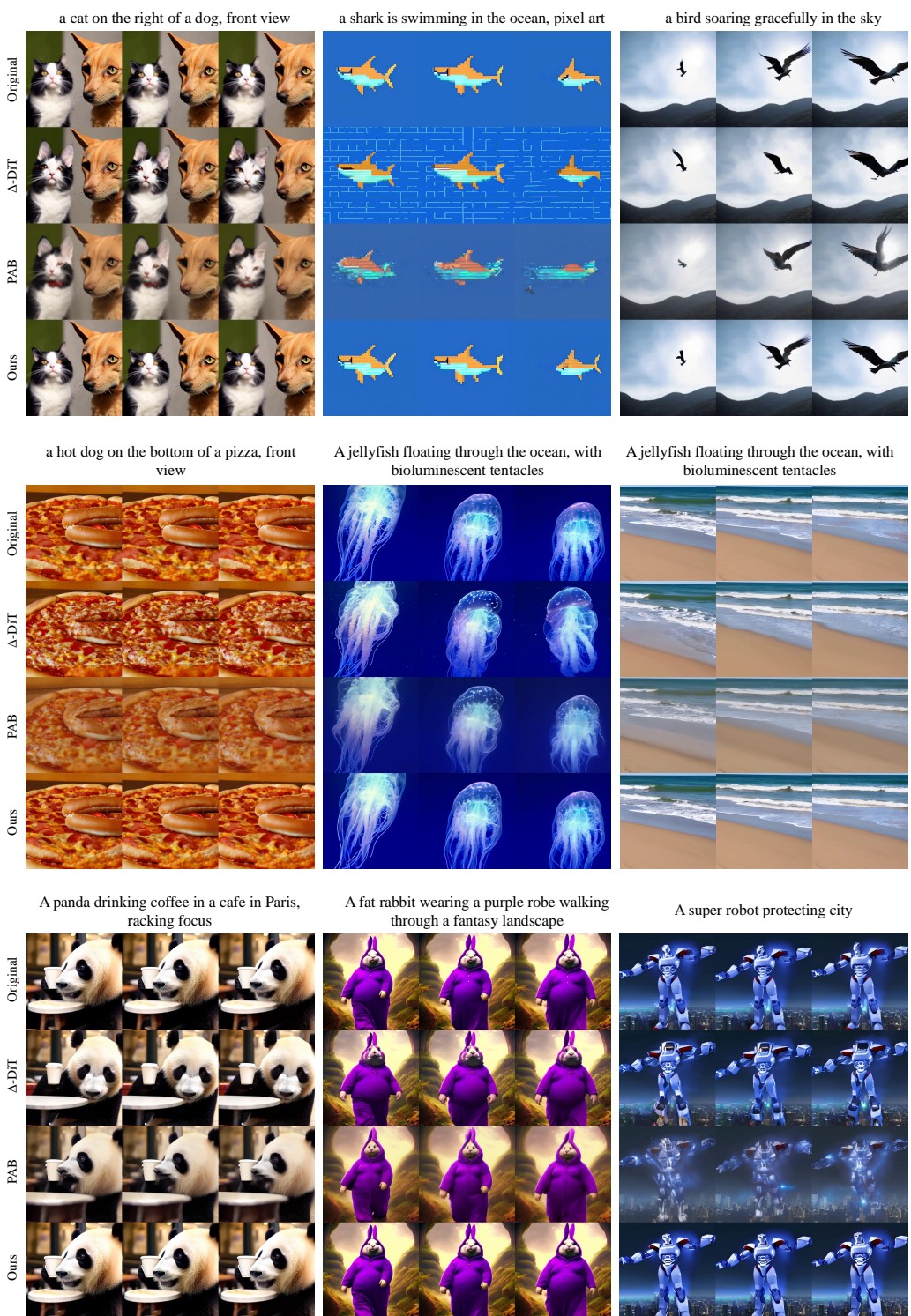

Figure 19: More visual results on Latte (512×512 16 frames). Zoom in for details.

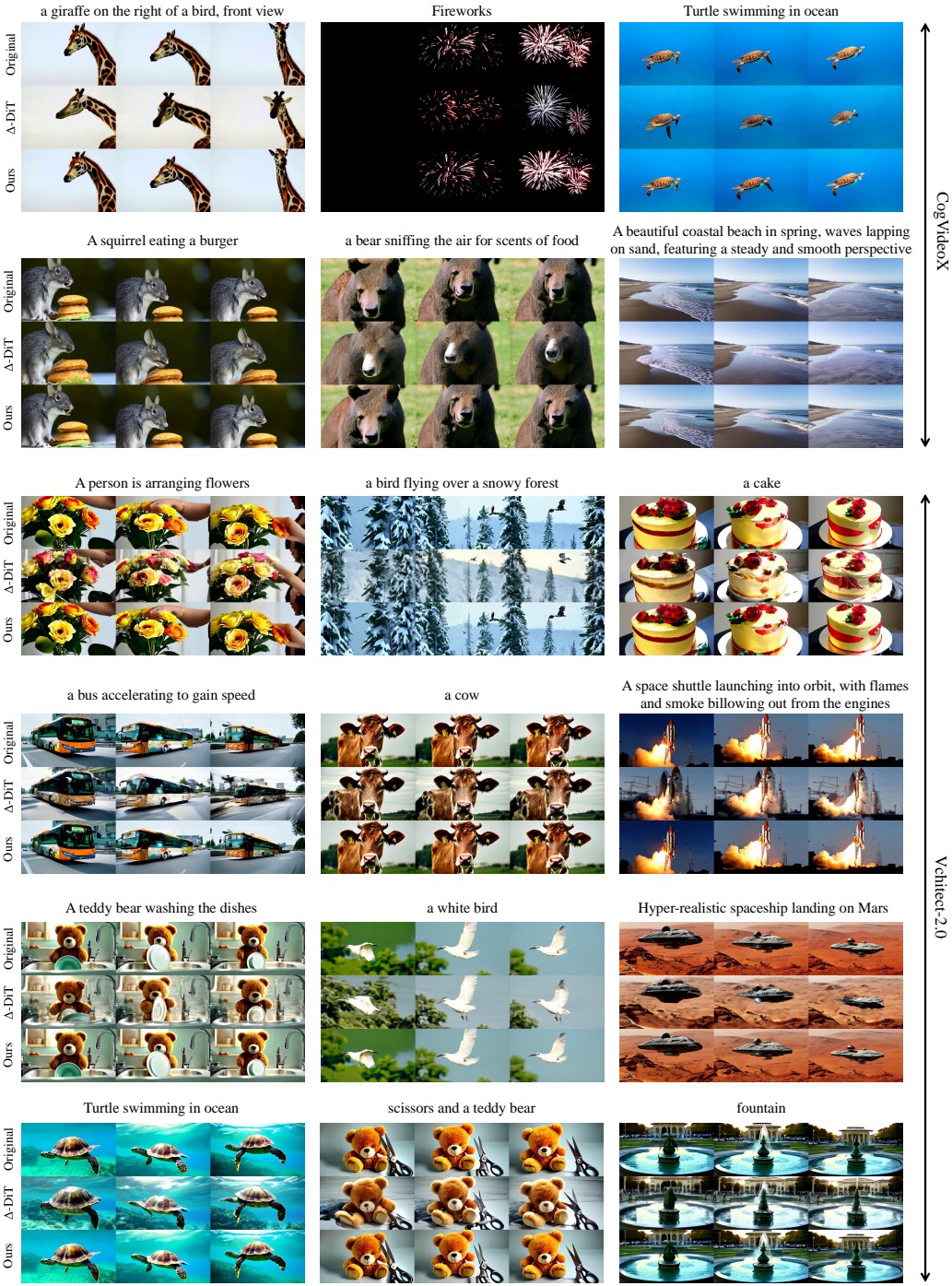

Figure 20: More visual results on CogVideoX-2B (480P 48 frames) & Vchitect-2.0 (480P 40frames).

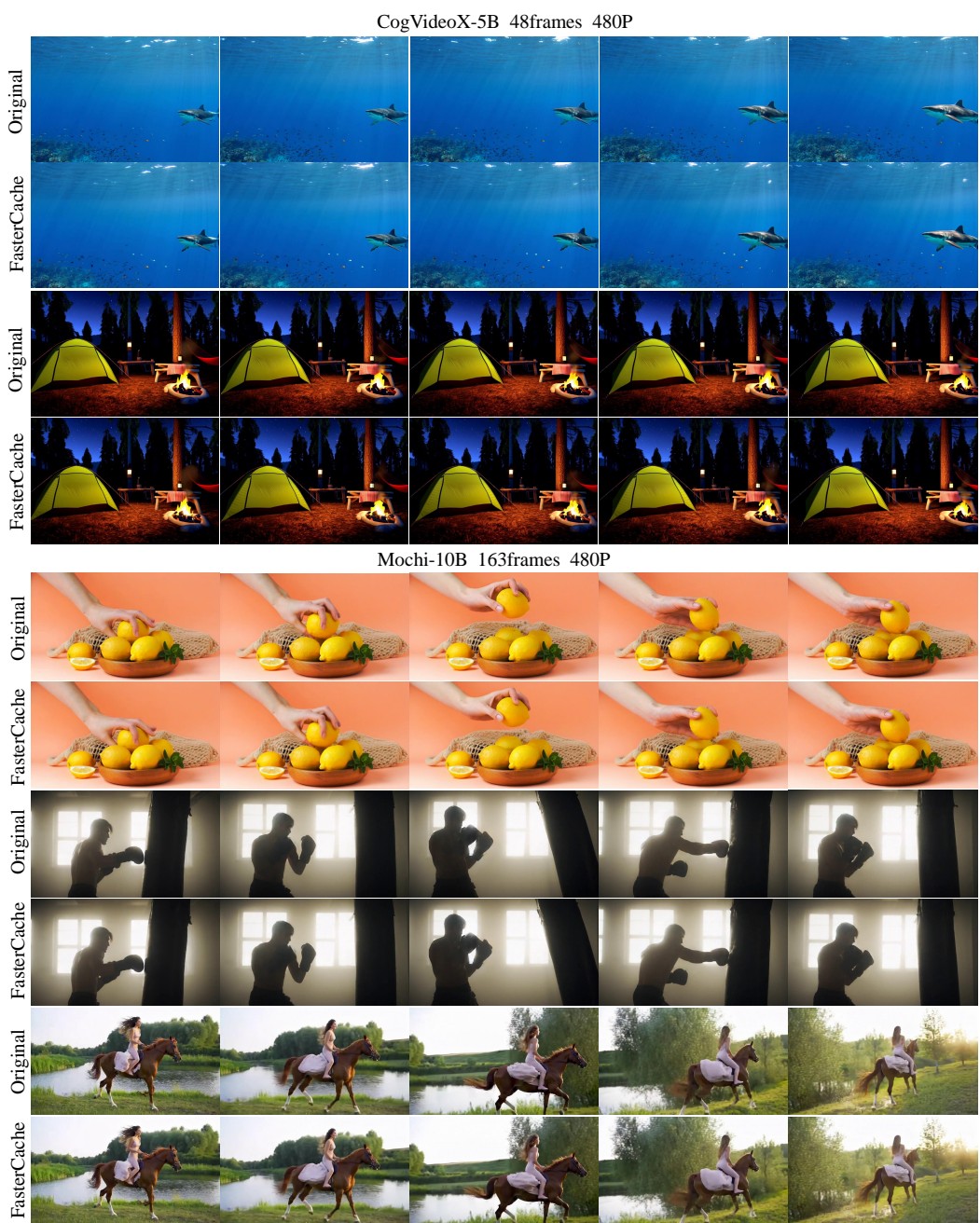

Figure 21: More visual results on CogVideoX-5B and Mochi-10B. Zoom in for details.

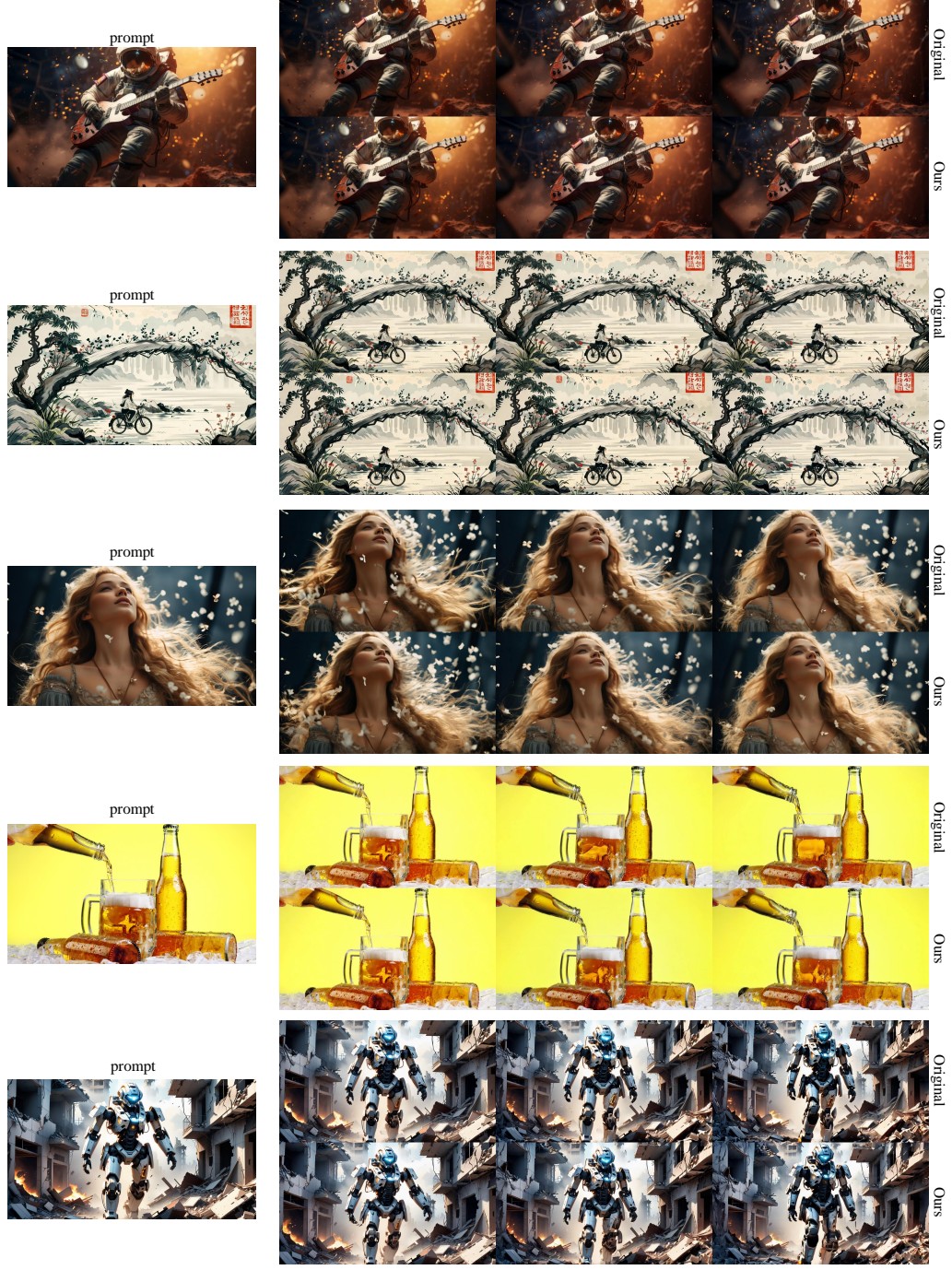

Figure 22: More visual results on DynamiCrafter (1024×576 16frames). Zoom in for details.

