# OpenReview forum: "FasterCache: Training-Free Video Diffusion Model Acceleration with High Quality"
_ICLR.cc/2025/Conference — ICLR 2025 Poster_

### Official Review · Reviewer_zNbc · 2024-10-31

**Soundness:** 3
**Presentation:** 3
**Contribution:** 2
**Rating:** 5
**Confidence:** 4

**Summary:**

This paper presents a method to accelerate the computation for video diffusion models. Based on the observation that adjacent-time features exhibit subtle differences,  the authors propose a dynamic feature reuse scheme. Based on the observation that the conditional and unconditional generation features have strong similarity in CFG, the authors propose a CFG-Cache scheme to save computations of unconditional stage.

**Strengths:**

this paper is relatively well organized and written.

**Weaknesses:**

this paper lacks novelty and insight. the proposed method is not solid in terms of logical and theoretical justification. the speedup of computation could be achieved using other methods.

**Questions:**

1. what is the reason of using equation (5) for feature reuse? it seems to be quite heuristic.
2. why in cfg, the similarity of two stages has such a wired characteristics (time-varing low and high frequency similarity)? there should be more analysis and insight.
3. in terms of saving the computation of unconditional stage, how about existing and well-adopted methods such as "On Distillation of Guided Diffusion Models"?
4. how does your method work with video generation models using flow matching?

---

> ### Author Response · Authors · 2024-11-20
> **Response to Reviewer zNbc (part 1)**
>
> Thank you for your review! We address the concerns below:
>
> ***W1: Lack of novelty and insight***
>
> (a) This paper is the first to provide a systematic analysis of existing cache-based methods. Previous approaches have primarily focused on reusing features in attention modules without deeply exploring reuse strategies. We are the first to identify that directly reusing features can lead to a loss of video details, as subtle differences between adjacent-step features in attention modules, if ignored, can degrade output quality.
>
>
>
> (b) We introduce the novel concept of dynamic feature reuse, which maintains high-quality generation while accelerating inference.
>
>
>
> (c) We are also the first to analyze feature redundancy in classifier-free guidance (CFG) and propose CFG-Cache, an training-free acceleration strategy specifically for CFG.
>
>
>
> Overall, through our in-depth analysis of cache-based methods and the introduction of effective solutions, we have significantly advanced the real-world application of cache-based methods. FasterCache has already been successfully implemented in various video generation models, achieving high-quality acceleration, including in the latest CogVideoX-5B and Mochi-10B.
>
> ***W2: Not solid in terms of logical and theoretical justification.***
>
>
>
> Thank you for your suggestion. We have added further discussion on the logical and theoretical justification for FasterCache designs in Appendix A.3 and Appendix A.5.
>
>
>
> Recently, the feasibility of cache-based methods for sampling acceleration in generative models has been widely validated[r1,r2,r3,r4,r5]. The phenomenon of high similarity between features at adjacent sampling time steps, which has been widely observed, allows cache based methods to achieve sampling acceleration by caching and reusing features from adjacent timesteps. FasterCache rethinks the issues with previous feature reuse methods and introduces the Dynamic Feature Reuse Strategy to maintain high-quality details during acceleration. Furthermore, FasterCache identifies significant redundancy in CFG computation and introduces CFG-Cache for further acceleration.
>
> **(1) Motivation for Dynamic Feature Reuse.** In Section 2.2, we discussed the motivation behind the additional biases term introduced in Eq.(5) in Dynamic Feature Reuse: Simply reusing attention features while ignoring their differences across adjacent timesteps can lead to detail loss. We also provided more discussion and experiments about the design choices in the Appendix A.3.1.
>
>
>
> **(2) Motivation for CFG-Cache.** In Section 2.3, we explained the introduction for CFG-Cache: the difference between conditional and unconditional outputs at the same time step is relatively small during the later stages of sampling. Additionally, we presented the motivation for adaptive enhancement of high- and low-frequency biases in Section 2.4 and Fig. 7. We also provided more discussion and visualization in Appendix A.3.2.
>
>
>
> **(3) Hyperparameters and design choices.** In Section 3.3 and Appendix A.3 and A.5, we also discussed and validated the effectiveness of hyperparameters and design choices through ablation study and comparison experiments. Please kindly refer to these sections for further details. Thank you!
>
>
>
> [r1] Ma, Xinyin, Gongfan Fang, and Xinchao Wang. "Deepcache: Accelerating diffusion models for free." CVPR. 2024.
>
>
>
> [r2] Li, Senmao, et al. "Faster diffusion: Rethinking the role of unet encoder in diffusion models." arXiv e-prints (2023): arXiv-2312.
>
>
>
> [r3] Chen, Pengtao, et al. "$\Delta $-DiT: A Training-Free Acceleration Method Tailored for Diffusion Transformers." arXiv preprint arXiv:2406.01125 (2024).
>
>
>
> [r4] Zhao, Xuanlei, et al. "Real-time video generation with pyramid attention broadcast." arXiv preprint arXiv:2408.12588 (2024).
>
>
>
> [r5] Kahatapitiya, Kumara, et al. "Adaptive Caching for Faster Video Generation with Diffusion Transformers." arXiv preprint arXiv:2411.02397 (2024).

---

> > ### Comment · Reviewer_zNbc · 2024-11-23
> >
> > On Motivation for Dynamic Feature Reuse: In Appendix A.3.2, there is a strong assumption "When ∆t is sufficiently small, the second-order term becomes negligible", which is quite strong. Basically you are assuming that the signal change between adjacent time steps is linear, which lacks evidence in theory and statistics.
> >
> > On Motivation for CFG-Cache: In Appendix A.3.2, the same strong assumption "we find that when ∆t is sufficiently small, the ϵ can be considered negligible" should be justified. In Figure 14, that specific example indicates that the conditional and unconditional outputs themselves are high-frequency signals at later stages of diffusion. So I wonder does it necessary to add a time-varying weighting mechanism? I have not seem a ablation study on this issue.

---

> > > ### Author Response · Authors · 2024-11-24
> > >
> > > * **The justification for the design of the error estimation bias in DFR.**
> > >
> > > Thank you for your suggestion. We have provided additional explanations regarding this issue in Appendix A.3.1 and Fig. 13. Based on the statistics of approximately 200 video samples, we plotted the magnitudes of the first-order and second-order terms of $F(t)$.  It can be observed that when $\Delta t$ is sufficiently small (e.g. $\Delta t=1$), **the second-order term is significantly smaller than the first-order term**, as shown in Fig. 13 (c). Furthermore, we tested three different estimations for $F(t)$, denoted as $\hat{F}(t)$: (a)$\hat{F}(t)=F(t+1)$, (b) $\hat{F}(t)=F(t+1) - \frac{dF(t)}{dt}$,  and (c) $\hat{F}(t)=F(t+1) - \frac{dF(t)}{dt} - \frac{d^2F(t)}{2dt^2}$. Subsequently, we calculated the L1 distance between each $\hat{F}(t)$ and $F(t)$. As shown in Fig. 13 (d), the inclusion of **the second-order term results in a marginal reduction in the L1 distance compared to the inclusion of the first-order term**. Considering the **memory and computational costs**, we use only the first-order term for error estimation in Dynamic Feature Reuse. Extensive experiments demonstrate that our first-order error estimation biases achieves a better trade-off between efficiency and visual quality compared to Vanilla Feature Reuse. In future work, we will continue to explore and investigate other potential designs for greater accuracy. Thank you.
> > >
> > > * **CFG-Cache and time-varying weighting mechanism.**
> > >
> > > Thank you for recognizing the motivation behind CFG-Cache. As shown in Fig. 14, in the later stages of sampling, the CFG primarily contributes to the synthesis of high-frequency details in the output. And the time-varying weighting mechanism helps focus more on detail synthesis during the later stages of synthesis. The discussion of the effectiveness of time-varying weighting mechanism could be found in Table 3 and Figure 9 in Section 3.3. Enhancing the high-frequency biases in the later stages could help improve the detail quality in the synthesized video.

---

> > > > ### Comment · Reviewer_zNbc · 2024-11-25
> > > >
> > > > In Fig. 13 (c), the second-order component is of range around 0.01, while the first-order component is of range around 0.03, so that the second-order term is not a term that can be considered negligible. Moreover, in Fig. 13 (d), the authors only compared the approximation error but not the ultimate generation quality. After all, a smaller term in approximation could have a large impact on the performance of generation.
> > > >
> > > > In Table 3, does the performance of "CFG-Cache w/o Enhancement" include FR? Is it unfair to compare "Vanilla FR" and "Full (w/ Dynamic FR) " as the later include CFG-Cache?

---

> > > > > ### Author Response · Authors · 2024-11-25
> > > > >
> > > > > (1)  Thank you for your insightful comments. We have refined the explanation of the second-order terms in Appendix A.3.1 to better clarify the motivation behind Dynamic Feature Reuse. The FasterCache framework inherently supports the incorporation of higher-order terms if desired by the user.  **Introducing second-order terms marginally reduces visual errors but comes at a significant cost of increased inference memory and latency**. For instance, when synthesizing videos at 360P with 48 frames on Open-Sora, incorporating second-order terms leads to a 20% increase in memory consumption compared to first-order error estimation, while the improvement in visual quality is not particularly significant and the ultimate visual quality metrics is detailed in the table below. **From the perspective of simplicity and efficiency**, FasterCache currently only employ first-order estimation. In future work, we plan to further investigate more efficient and accurate estimation designs.
> > > > >
> > > > > |  | VBench | LPIPS | PSNR | SSIM |
> > > > > |----------|----------|----------|----------|----------|
> > > > > | Original    | 79.06% | -  | -   | -|
> > > > > | Vanilla Feature Reusing | 78.31% | 0.0662 | 27.98 |0.8832|
> > > > > | Dynamic Feature Reusing (first order) | 78.77% | 0.0601 | 28.30 |0.8974|
> > > > > | Dynamic Feature Reusing (second order) | 78.80% | 0.0590 | 28.34 |0.8986|
> > > > >
> > > > > (2) Thank you for your suggestion. Table 3 includes three groups of experiments. The first group presents the performance of the Original Open-Sora. The second group shows the results of using Vanilla FR and Dynamic FR within the full FasterCache framework (both with CFG-Cache), validating the effectiveness of Dynamic FR. The third group provides results with different CFG-Cache enhancement settings within the full FasterCache framework, demonstrating the effectiveness of the time-varying enhancement. The experimental setup for each group ensures a fair basis for drawing conclusions. We will include more explanation of this part in the revised version to avoid potential ambiguity.

---

> > > > > > ### Comment · Reviewer_zNbc · 2024-12-02
> > > > > >
> > > > > > Thanks for the extra experiments and explanation on Table 3. I think the current writing in Table 3 is misleading (e.g. the second group does not indicate using the CFG-Cache). My concerns on second-order approximation is mostly addressed, but I still think the degree of novelty is limited. I will raise my score accordingly.

---

> > > > > > > ### Author Response · Authors · 2024-12-02
> > > > > > >
> > > > > > > We sincerely appreciate the reviewer's insightful comments and recognition of the motivation of the bias term in Dynamic Feature Reuse. Additionally, we will revise the writing in Table 3 to avoid potential misunderstandings. Thank you for your valuable suggestions!

---

> ### Author Response · Authors · 2024-11-20
> **Response to Reviewer zNbc (part 2)**
>
> ***W3: The speedup of computation could be achieved using other methods.***
>
>
>
> **(1) Distillation and pruning based methods.** In the field of acceleration, techniques such as distillation and pruning have been explored. However, distillation requires **substantial data and computational resources**, while pruning inevitably leads to quality degradation. As a result, the application of these methods in video synthesis acceleration remains relatively limited.
>
>
>
> **(2) Cache based methods.** The cache based method, due to its **training-free and plug-and-play properties**, has been widely applied and studied. For instance, Delta-DiT [r1], which utilizes a residual caching mechanism between attention layers, PAB [r2], which is based on an attention feature broadcasting mechanism, and the recent AdaCache [r3] from Meta with adaptive caching. The feasibility of cache based methods has been widely validated and is gradually becoming one of the mainstream acceleration approaches for video synthesis acceleration.
>
>
>
> **(3) The efficiency and effectiveness of FasterCache.** Through observing feature variations, we identified subtle differences overlooked in Vanilla Feature Reuse, leading to the introduction of Dynamic Feature Reuse to capture these variations. Additionally, we discovered significant redundancy in the computation of CFG, which prompted the introduction of CFG-Cache for acceleration. By combining Dynamic Feature Reuse and CFG-Cache, FasterCache achieves a favorable trade-off between efficiency and effectiveness. For example, on Vchitect 2.0, FasterCache achieved **80.84% VBench score and 1.67$\times$ speedup**, significantly outperforming the second-best result of **79.56% VBench score and 1.26$\times$ speedup**.
>
> [r1] Chen, Pengtao, et al. "$\Delta $-DiT: A Training-Free Acceleration Method Tailored for Diffusion Transformers." arXiv preprint arXiv:2406.01125 (2024).
>
>
>
> [r2] Zhao, Xuanlei, et al. "Real-time video generation with pyramid attention broadcast." arXiv preprint arXiv:2408.12588 (2024).
>
>
>
> [r3] Kahatapitiya, Kumara, et al. "Adaptive Caching for Faster Video Generation with Diffusion Transformers." arXiv preprint arXiv:2411.02397 (2024).
>
>
> ***Q1: what is the reason of using equation (5) for feature reuse?***
>
> Thank you for your valuable suggestion. We have added more motivation for Dynamic Feature Reuse in Section 2.4 and Appendix A.3.1.
>
> As discussed in Section 2.2, we note that while the attention features between adjacent timesteps are highly similar, there are still noticeable differences between them, which remain relatively stable (Fig. 5, right). Simply reusing previous features while ignoring these differences can result in a loss of detail (Fig. 5, left). To address this, we leverage these differences to construct a bias term (Eq. (5)) that estimates the trend of feature variation, enabling accurate capture of evolving details across timesteps. This bias term **essentially estimates the error between the cached feature $F^t_{cache}$ from the previous timestep and the current feature $F_{t-1}$**, as discussed in Appendix A.3.1.
>
> For the choice of $w(t)$ in Eq. (5), we have provided additional experiments with different design options and found that linearly increasing $w(t)$ is a simple yet effective design. Please kindly refer to Appendix A.3.1 for further details. Thank you.
>
>
> ***Q2: why in cfg, the similarity of two stages has such a wired characteristics (time-varing low and high frequency similarity)? there should be more analysis and insight.***
>
> Thank you for your insights. We have added further explanations about CFG-Cache in Section 2.4 and Appendix A.3.2.
>
> From the onset of CFG-Cache to the end of sampling process, the differences between the conditional and unconditional output features progressively shift from being dominated by low-frequency features to high-frequency features. This observation aligns with the feature visualization analysis in Fig. 15: during the early and middle stages of sampling, CFG primarily guides the model to **synthesize perceptual features such as reasonable shapes and layouts**, which are often represented in the low-frequency domain. In contrast, during the later stages of sampling, CFG **contributes primarily to the synthesis of high-quality details**, typically governed by high-frequency features. Studies on distillation-based methods aimed at reducing computations in unconditional stages [r1] has also observed similar phenomena. This insight motivates us to assign higher weights to features of different frequencies at different stages to enhance focus and reduce interference, thereby preserving the visual quality of the synthesized results.
>
> [r1] Hsiao, Yi-Ting, et al. "Plug-and-Play Diffusion Distillation." CVPR. 2024.

---

> > ### Comment · Reviewer_zNbc · 2024-11-23
> >
> > "there are still noticeable differences between them, which remain relatively stable (Fig. 5, right).": this is just one specific example (moreover, without numerical measurement of similary), which cannot lead to a solid justification.
> >
> > "This bias term essentially estimates the error between the cached feature from the previous timestep and the current feature, as discussed in Appendix A.3.1." : similar to my previous comment, the key assumption that quadratic and higher-order terms can be omitted is strong, and there is no analysis on the error bound and optimality of this estimator.

---

> > > ### Author Response · Authors · 2024-11-24
> > >
> > > * **The statistical basis for the rate of change of the first-order biases.**
> > >
> > > Thank you for your suggestion. We have incorporated the results obtained from the statistics of approximately 200 videos. From Fig. 13 (c), it can be observed that the norm of the second-order term is significantly smaller than that of the first-order term, indicating the slow variation of biases of features between adjacent time steps. Moreover, the experiments in the ablation study in Section 3.3 statistically demonstrate the effectiveness of Dynamic Feature Reuse.

---

> > > > ### Comment · Reviewer_zNbc · 2024-11-25
> > > >
> > > > My concerns remain unresolved. Please refer to my previous comment.

---

> ### Author Response · Authors · 2024-11-20
> **Response to Reviewer zNbc (part 3)**
>
> ***Q3: in terms of saving the computation of unconditional stage, how about existing and well-adopted methods such as "On Distillation of Guided Diffusion Models"?***
>
> For image synthesis, distillation techniques [r1,r2,r3] enables a reduction in the number of sampling steps required and eliminates the computational cost of the unconditional branch. However, **the application of distillation in video synthesis remains limited**, possibly due to the substantial computational and data resources required for distilling pre-trained models (for example, [r4] utilizes 64 A100 GPUs), and the inherent complexity of video synthesis tasks. Our method offers a training-free approach that significantly reduces the computational cost while preserving the fidelity of the synthesized videos. In future work, we plan to explore the integration of our method with distillation-based approaches. We sincerely appreciate your suggestions.
>
> [r1] Meng, Chenlin, et al. "On distillation of guided diffusion models." CVPR. 2023.
>
> [r2] Song, Yang, et al. "Consistency models." arXiv preprint arXiv:2303.01469 (2023).
>
> [r3] Luo, Simian, et al. "Latent consistency models: Synthesizing high-resolution images with few-step inference." arXiv preprint arXiv:2310.04378 (2023).
>
> [r4] Lin, Shanchuan, and Xiao Yang. "Animatediff-lightning: Cross-model diffusion distillation." arXiv preprint arXiv:2403.12706 (2024).
>
> ***Q4: how does your method work with video generation models using flow matching?***
>
> **Our method is applicable to video generation models using flow matching.** The core of our acceleration mechanism lies in reusing features during iterative denoising process (Dynamic Feature Reuse) and reducing the computational load of CFG (CFG-Cache). FasterCache has already been successfully implemented in various video generation models, achieving high-quality acceleration. For example, the Open-Sora 1.2 base model used in our experiments is trained with flow matching. Moreover, FasterCache is applicable to models of different architectures and scales, such as CogVideoX-2B, CogVideoX-5B, and Mochi-10B.
>
> ---
> We deeply appreciate the reviewer's efforts in helping make our paper more clear and stronger.
>
> **If there are any further questions, we are happy to dicuss them!**

---

> ### Author Response · Authors · 2024-11-23
> **Follow-up**
>
> Could you kindly confirm if our response has addressed your concerns? Please don’t hesitate to ask any questions or continue the discussion.
>
> Thank you!

---

### Official Review · Reviewer_JHfK · 2024-11-02

**Soundness:** 3
**Presentation:** 3
**Contribution:** 3
**Rating:** 6
**Confidence:** 3

**Summary:**

This paper propose a training-free method for efficient video generation. Their method is based on the attention feature map and output redundancy observed in the forward pass of the denoising process. To this end, they propose to (1) use the feature map dynamics to restore the missing details in the previous naive feature map reuse method. (2) use the frequency difference to compensate the cached conditional output in CFG. The proposed method has been validated with several models and showed promising results.

**Strengths:**

- The paper is well-written and easy to read.
- The proposed attention feature map dynamics could be a generic method and applied into all other cache based efficient generation methods.
- the proposed frequency residual method for CFG is novel and seems also a generic method that can be applied into CFG for all diffusion based image/video generation models.

**Weaknesses:**

- Distillation based efficient generation methods are less discussed and not added into the comparison. I can understand that this work mainly focused on the cache based method. But:
  - What's the delta of this method compared to distillation based method?
  - Are they compatible to each other? If not, in real world applications, which method we need to use?

- the overall method seems to perform better than all other methods. What's the contribution of each individual method? What the cost of additional computation in each individual method?

**Questions:**

Please refer to the weakness section.

---

> ### Author Response · Authors · 2024-11-20
> **Response to Reviewer JHfK**
>
> Thank you for your review! We address the concerns below:
>
> ***Q1: What's the delta of this method compared to distillation based method?***
>
> Our approach differs from distillation-based methods in terms of **acceleration mechanisms** and **implementation costs**.
>
> **(1) Acceleration Mechanism.** Most existing distillation-based methods achieve efficient synthesis by fine-tuning the pre-trained models to reduce the number of sampling steps required. Our approach accelerates sampling in a training-free manner by analyzing the feature variations during the sampling process, properly caching and reusing these features to reduce redundant computation.
>
> **(2) Implementation Costs.** Distillation-based methods typically require **significant computational resources** and **a vast amount of high-quality data** for fine-tuning the pre-trained models. Furthermore, when the model undergoes iterative updates, previously distilled results are difficult to reuse. Our method requires **no additional training and is plug-and-play**. It is compatible with most frameworks that employ classifier-free guidance and iterative denoising generation, remaining unaffected by model updates. Therefore, cache-based acceleration method is a more versatile and ready-to-use solution.
>
> Thank you for your suggestion. We will include a more detailed discussion in the related work.
>
>
>
> ***Q2: Are they compatible to each other?***
>
> **(1) Compatibility.** Since the acceleration mechanisms of distillation-based methods and cache-based methods are orthogonal, they are theoretically compatible. However, research on distillation techniques for video synthesis models remains highly limited. In future work, we will explore combining distillation-based and cache-based methods for faster acceleration method.
>
> **(2) Selections for real-world applications.** In video synthesis acceleration, cache-based methods have been widely studied and adopted for their training-free and plug-and-play properties. On the other hand, distillation-based acceleration techniques, due to their requirements for substantial computational and data resources, may be more suitable for scenarios with ample resources. Possibly due to the resource requirements and the complexity of video synthesis task, the exploration of video distillation models is still very limited at present.
>
>
>
> ***Q3: What's the contribution of each individual method? What the cost of additional computation in each individual method?***
>
> Thank you for your insights. The overall methodology of FasterCache integrates two acceleration components: **Dynamic Feature Reuse (DFR)** and **CFG-Cache.**
>
> **(1) Dynamic Feature Reuse (DFR).** **(a) Contribution:** DFR focuses on reusing features between adjacent sample steps within the same attention layer. We observed that Vanilla Feature Reuse tends to cause detail loss. To mitigate this, we introduced an additional bias term, aiming to accelerate inference while minimizing quality degradation. **(b) Additional Computation:** The computational cost of DFR is almost the same as that of vanilla Feature Reuse. As shown in Table 2, the inference time of Vanilla Feature Reuse is 33.25s, while DFR requires 33.50s. While adding almost no computational overhead, DFR improves the VBench score from 78.34% to 78.69% (Table 3).
>
>
>
> **(2) CFG-Cache. (a) Contribution:** CFG-Cache primarily aims to reduce the computational cost of the unconditional stage. We observed that the difference between conditional and unconditional outputs at the same step is minor. Based on this, we designed CFG-Cache to cache the high- and low-frequency biases between conditional and unconditional outputs and adaptively enhance them during reuse, accelerating inference while maintaining video synthesis quality. **(b) Additional Computation:** CFG-Cache involves computing high- and low-frequency biases between conditional and unconditional outputs and enhancing them during reuse. The additional computation is significantly smaller than the saved computation cost, making CFG-Cache alone capable of substantially reducing inference latency(In Tab. 2, Original: 41.28s vs. w/ CFG-Cache: 31.32s).
>
>
> Since the two components accelerate the inference process from different aspects, they can be combined. Thus, our overall method, FasterCache, outperforms any individual method in terms of efficiency. Compared to the attention feature broadcasting mechanism of PAB and the residual caching mechanism between attention layers in Delta-DiT, our FasterCache is also capable of better accelerating the inference process while maintaining visual quality. For example, on Vchitect 2.0, FasterCache achieved **80.84% VBench score and 1.67$\times$ speedup**, significantly outperforming the second-best result of **79.56% VBench score and 1.26$\times$ speedup**.
>
> ---
> We deeply appreciate the reviewer's efforts in helping make our paper more clear and stronger.
>
> **If there are any further questions, we are happy to dicuss them!**

---

> ### Author Response · Authors · 2024-11-23
> **Follow-up**
>
> Thank you for reviewing our work and appreciating its value.
>
> If there are any questions or concerns, we are more than happy to discuss them further.
>
> Thank you very much.

---

> ### Author Response · Authors · 2024-12-03
> **Kind Reminder**
>
> Dear Reviewer JHfK,
>
> We sincerely appreciate the effort you have put into reviewing our submission. We have submitted our response along with new results and would like to follow up to confirm whether our response has addressed your concerns.
>
> If you have any remaining questions, please feel free to let us know. We look forward to your feedback and are prepared to address any issues promptly before the discussion deadline.
>
> Thank you once again!

---

### Official Review · Reviewer_4a6E · 2024-11-04

**Soundness:** 2
**Presentation:** 3
**Contribution:** 3
**Rating:** 6
**Confidence:** 4

**Summary:**

This paper presents an innovative approach to accelerate video diffusion models without sacrificing quality. By analyzing existing cache - based methods and classifier - free guidance (CFG), the authors identify limitations and redundancies. They introduce FasterCache, which includes a dynamic feature reuse strategy for attention modules and CFG-Cache. The dynamic feature reuse strategy adjusts features across timesteps, maintaining distinction and continuity. CFG-Cache optimizes the reuse of conditional and unconditional outputs. Experimental results on multiple video diffusion models demonstrate significant acceleration while keeping video quality comparable to the baseline, outperforming existing methods.

**Strengths:**

In terms of originality, the proposed FasterCache method is highly innovative. It combines a dynamic feature reuse strategy that considers subtle differences between adjacent timesteps and a CFG - Cache component for handling conditional and unconditional outputs in a novel way. This is a departure from existing methods and reveals new acceleration opportunities through a pioneering investigation of classifier - free guidance (CFG).

Regarding quality, a thorough analysis of existing cache - based methods and CFG is provided. This includes identifying problems and limitations, and conducting in - depth investigations into feature reuse and CFG outputs, which forms a solid foundation for the method. The experimental design is robust, with testing on multiple video diffusion models and the use of various evaluation metrics. Ablation studies further demonstrate the effectiveness of each component.

For clarity, the paper has a clear structure, starting with an introduction to the problem and existing solutions, followed by a detailed description of the method in the methodology section. The experimental results are presented clearly, and the discussion and conclusions are well - organized. Complex concepts are explained accessibly, such as through diagrams and detailed descriptions of the key components.

In terms of significance, the research has practical implications as it can reduce the time and computational resources required for video generation, which is important for applications like video content creation, virtual reality, and augmented reality. It also advances the field by filling gaps in existing research on video diffusion models, addressing limitations of current acceleration methods, and inspiring further studies on improving efficiency and performance.

**Weaknesses:**

Hyperparameter Explanation. Some of the hyperparameters used in the method, such as the weighting function are not explained in sufficient detail. Although default values are provided and it is mentioned that they work well for most models, a more in-depth discussion on how these hyperparameters are chosen and their impact on the performance for different datasets and models would make the method more reproducible and understandable.

Uncertainty in complex scenes. The paper mentions that in complex scenes with substantial video motion, the method may occasionally produce degraded results. This may limit the practical applicability of the method once video generation model baselines are imporved and complex video content is common. More discussions refer to the Question 1.

Weak theoretical underpinning: The paper focuses more on the empirical evaluation and practical implementation of the FasterCache method. Although the experimental results are strong, the theoretical foundation behind some of the proposed techniques could be strengthened. For example, a more in-depth analysis of why the dynamic feature reuse strategy and CFG-Cache work as expected in different scenarios from a theoretical perspective would add more credibility to the method.

**Questions:**

Questions:
1. Is the effectiveness of the Dynamic Feature Reuse Strategy because the videos generated by existing video generation methods are relatively smooth, with little difference between adjacent frames, e.g. fixed backgrounds? If so, once the motion of the video is large and the adjacent frames change greatly, then is the proposed method still effective?

2. Negative prompt will affect the quality of video generation, which generally improves it. So will CFG-Cache and the separation of high and low frequencies have an impact on negative prompt setting?

3. For Open-Sora-Plan, although FasterCache achieved an acceleration of 1.68x, the generated video has significant varying in pixel domain compared to the original one (PSNR 23.72). What is the reason for this bias? Blurriness, noise, or changes in semantic level?

4. Will the setting of g value in CFG formula 4 (Line 167) affect the performance of FasterCache?

---

> ### Author Response · Authors · 2024-11-20
> **Response to Reviewer 4a6E (part 1)**
>
> Thank you for your review! We address the concerns below:
>
> ***W1: Hyperparameter Explanation. Some of the hyperparameters used in the method, such as the weighting function are not explained in sufficient detail.***
>
>
>
> Thank you very much for your suggestion. We have added additional experiments on hyperparameters
> and design choices to the Appendix A.3.1 and A.5.
>
>
>
> **(1) weighting function $w(t)$.**  We compared three designs for the weighting function $w(t)$: constant $w(t)$, optimized $w(t)$, and the linearly increasing $w(t)$ that we ultimately adopted. As shown in Fig. 13 and Table 5 (shown below), the results show that: (1) The optimized $w(t)$ performed best, with Linearly increasing achieving comparable results. (2) The Learnable weights $w(t)$ obtained through optimization, gradually increase as sampling progresses. Considering **effectiveness and simplicity**, we ultimately chose the linearly interpolated $w(t)$. For more details and results, please kindly refer to Appendix A.3.1. Thank you.
>
> | Method                     | LPIPS  | PSNR  | SSIM   |
> |----------------------------|--------|-------|--------|
> | Vanilla FR                | 0.0657 | 28.20 | 0.8785 |
> | Dynamic FR (w(t)=0.5)     | 0.0615 | 28.33 | 0.8889 |
> | Dynamic FR (learned w(t)) | 0.0596 | **28.45** | **0.8941** |
> | Dynamic FR (linear w(t))  | **0.0590** | 28.41 | 0.8938 |
>
> **(2) Different caching timestep interval of Dynamic Feature Reuse.** In Appendix A.5.2, we experimented with different caching intervals of Dynamic Feature Reuse. We can adjust the the caching timestep interval based on acceleration needs and fidelity requirements.
>
> **(3) Different caching timestep interval of CFG-Cache.**  In Appendix A.5.2, we experimented with different CFG-Cache intervals and found that when the interval exceeds 5 timesteps, there is a significant decline in fidelity. Therefore, to balance fidelity and efficiency, we chose a CFG-Cache interval of 5. This means that after CFG-Cache starts to take effect,, the model performs full inference for both the conditional and unconditional branches every 5 timesteps and caches the feature biases.
>
> **(4) Different settings of $\alpha$ in CFG-Cache** In Appendix A.5.2, we experimented with different $\alpha$ settings in CFG-Cache. And we found that $\alpha_1=\alpha_2=0.2$ works effectively.
>
>
> ***W2: Uncertainty in complex scenes.***
>
> **(a)** We observed that the uncertainty of complex scenes only appeared in certain models, such as Open-Sora and Open-Sora-Plan. In recent experiments, we found that implementing FasterCache on other models, such as CogVideoX-2B, CogVideoX-5B, and Mochi-10B, maintained high quality and fidelity even in scenarios with intense motion or intricate scenes. **The new results have been added in Fig. 21 in Appendix A.4.**
>
> **(b)** Based on this observation, we hypothesize that this is due to the capability of the base models. Models like CogVideoX-2B, CogVideoX-5B, and Mochi-10B have strong generation abilities that can handle complex spatiotemporal features. Thus, even when accelerated with FasterCache, they are still able to ensure high-quality generation for complex scenes.

---

> ### Author Response · Authors · 2024-11-20
> **Response to Reviewer 4a6E (part 2)**
>
> ***W3: Weak theoretical underpinning***
>
> Thank you for your suggestion. We have added a discussion on the effectiveness of Dynamic Feature Reuse and CFG-Cache in Appendix A.3.1 and Appendix A.3.2.
>
>
>
> **(1) Dynamic Feature Reuse.** Assume that the output features of a particular layer in the diffusion model are a function of the timestep $t$, denoted as $F(t)$. The motivation behind Vanilla Feature Reuse lies in the observation that features at adjacent timesteps are highly similar. Vanilla Feature Reuse avoids the computation at the current timestep by directly reusing the features from the previous timestep, i.e. $F(t) = F(t+\Delta t)$. Although $F(t)$ and $F(t+\Delta t)$ are very close with a minimal error $E=F(t)-F(t+\Delta t)$, the difference is not zero. To estimate this error, we assume that $F(t)$ is a smooth and differentiable function with respect to $t$, allowing us to perform a Taylor expansion, yielding:
> $$
> F(t+\Delta t) = F(t) + \frac{dF(t)}{dt}\Delta t + \frac{d^2F(t)}{dt^2}\frac{\Delta t^2}{2} + O(\Delta t^3),
> $$
> $$
> F(t+3\Delta t) = F(t) + 3\frac{dF(t)}{dt}\Delta t + 3\frac{d^2F(t)}{dt^2}\frac{\Delta t^2}{2} + O(\Delta t^3).
> $$
>
> By subtracting these expansions, we derive:
> $$
> F(t+\Delta t) - F(t+3\Delta t) = (\frac{dF(t)}{dt}\Delta t)\times(-2) + O(\Delta t^2),
> $$
>
> When $\Delta t$ is sufficiently small, the second-order term becomes negligible. Using this result, the error term can be expressed as:
>
> $$
> E = F(t) - F(t+\Delta t) \approx - \frac{dF(t)}{dt}\Delta t = (F(t+\Delta t) - F(t+3\Delta t)) * w.
> $$
>
> The scale factor $w$ is introduced to scale the bias term to approximate the error $E$. In Eq. (5), $E=F_{t-1}-F_{cache}^t\approx(F_{cache}^t-F_{cache}^{t+2})*w(t)$. By introducing this bias feature term, the information loss could be reduced, thereby improving the quality of the synthesis videos while maintaining computational efficiency.
>
>
>
> **(2) CFG-Cache.** The reliability of CFG-Cache stems from three key factors:
>
> **(a)** After the early stage $t_{early}$, the similarity between conditional output $cond(t)$ and unconditional output $uncond(t)$ at the same timestep $t$:
>
> $$
> uncond(t) = cond(t) + \Delta, when\ t>=t_{early}.
> $$
>
> **(b)** The predictability of biases between conditional and unconditional output from previous timesteps $t + \Delta t$, expressed as:
> $$
> \Delta =  uncond(t+\Delta t) - cond(t+\Delta t) = uncond(t) - cond(t) + \epsilon.
> $$
>
> In practice, we found that when $\Delta t$ is sufficiently small, the $\epsilon$ can be considered negligible. Then:
> $$
> uncond(t) \approx cond(t) + (uncond(t+\Delta t) - cond(t+\Delta t))
> $$
>
> **(c)** The dynamic variations of the frequency-domain distribution of feature biases, as shown in Fig. 7(b) and Fig.14.
>
>
> ***Q1: Is the effectiveness of the Dynamic Feature Reuse Strategy because the videos generated by existing video generation methods are relatively smooth, with little difference between adjacent frames, e.g. fixed backgrounds? If so, once the motion of the video is large and the adjacent frames change greatly, then is the proposed method still effective?***
>
>
>
> **(1) Mechanism and effectiveness of Dynamic Feature Reuse.** The effectiveness of the Dynamic Feature Reuse Strategy primarily derives from two properties of feature variation **along the sampling process steps**: (a) Within the same attention layer, features output at adjacent sampling steps are similar but not identical, neglecting these differences can lead to a loss of detail. (b) Although differences exist, these differences remain relatively stable over short sampling intervals, allowing us to construct a bias term in Eq.(5) within the Dynamic Feature Reuse Strategy to capture the evolving details across sampling steps.
>
>
>
> **(2) The relationship between video motion and Dynamic Feature Reuse.** Video motion and scene complexity are determined by **diversity and variations of tokens within the video token sequence.** Dynamic Feature Reuse, on the other hand, focuses on **how the same token changes across different sampling steps.** These are separate dimensions. Our Dynamic Feature Reuse Strategy works effectively for video generation across a range of scenes and motion levels. As shown in Fig. 21, our method successfully handles complex content and intense motion, such as in videos generated by CogVideoX-5B and Mochi-10B.

---

> ### Author Response · Authors · 2024-11-20
> **Response to Reviewer 4a6E (part 3)**
>
> ***Q2: Will CFG-Cache and the separation of high and low frequencies have an impact on negative prompt setting?***
>
> Thanks for the valuable suggestion.
>
> In Appendix A.3.3 and Fig. 15, we compared two different negative prompt settings on Open-Sora: (a) empty negative prompt and (b) non-empty negative prompt: “worst quality, normal quality, low quality, low res, blurry, text, watermark, logo, banner, extra digits, cropped, jpeg artifacts…”.
>
>
> **(1) Performance of different negative prompt settings.** We calculated the LPIPS, SSIM, and PSNR between the videos generated by FasterCache and those generated by the original model. As shown in Fig. 15, the experimental results show that **FasterCache exhibits similar performance under both prompt settings, consistently achieving high fidelity.** For example, under the default CFG scale ($g = 7 $) setting, the SSIM score of FasterCache with the empty negative prompt and the non-empty negative prompt are 0.8938 and 0.8916, respectively.
>
>
> **(2) Analysis of the impact of negative prompts setting.** The experimental results are consistent with our expectations, as CFG-Cache caches the **biases between the conditional and unconditional outputs**, which are not significantly affected by changes of the negative prompts.
>
>
>
> ***Q3: For Open-Sora-Plan, although FasterCache achieved an acceleration of 1.68x, the generated video has significant varying in pixel domain compared to the original one (PSNR 23.72). What is the reason for this bias? Blurriness, noise, or changes in semantic level?***
>
>
> These variations primarily arise from subtle semantic changes, likely due to the inherent sensitivity to feature variations of Open-Sora-Plan. Similarly, the comparative performance of the PAB and $\Delta-$DiT methods on Open-Sora-Plan is also weaker on the PSNR metric than their performance on Open-Sora.
>
>
> ***Q4: Will the setting of g value in CFG formula 4 (Line 167) affect the performance of FasterCache?***
>
>
> Thanks for the valuable suggestion. We added the analysis of the CFG guidance scale based on the Open-Sora model. Its default CFG scale is 7, and we experimented with CFG scales ranging from 2 to 12. As shown in Fig. 15 (b) and (c), regardless of increasing or decreasing the scale, while the adjustment affects the original Open-Sora results, **FasterCache consistently maintains a high level of alignment with the original results**, particularly in preserving details. Therefore, FasterCache is not affected by changes in the CFG guidance scale and maintains high-quality acceleration.
>
>
> ---
> We deeply appreciate the reviewer's efforts in helping make our paper more clear and stronger.
>
> **If there are any further questions, we are happy to dicuss them!**

---

> ### Author Response · Authors · 2024-11-23
> **Follow-up**
>
> Thank you once again for your thoughtful feedback and for recognizing the value of our work!
>
> If you have any additional questions or suggestions, we’d be delighted to discuss them further.
>
> Thank you very much!

---

> > ### Comment · Reviewer_4a6E · 2024-12-02
> > **Response to Authors**
> >
> > Thank you for your detailed response to my comments. Your responses appear to address the concerns raised, and I appreciate the clarifications and additional results provided.

---

> > > ### Author Response · Authors · 2024-12-02
> > > **Thank you!**
> > >
> > > Thank you so much! We are very glad that your concerns have been addressed!

---

### Official Review · Reviewer_4sX2 · 2024-11-04

**Soundness:** 3
**Presentation:** 3
**Contribution:** 3
**Rating:** 5
**Confidence:** 4

**Summary:**

This paper proposes a training-free feature caching strategy for video diffusion models, named FasterCache, that improves how features are reused for accelerating inference. Instead of directly reusing the adjacent-step features as in previous works, FasterCache interpolates the neighboring features to efficiently synthesize features for the missing steps. In addition, the paper introduces CFG-Cache, which aims to effectively reuse the conditional outputs for computing the unconditional outputs. Experimental results demonstrate notable visual quality and efficiency improvements compared with the existing methods.

**Strengths:**

1. The paper clearly demonstrates the motivation and the problem of existing works with illustrative examples (Fig. 3-7). Overall, the limitations of vanilla cache-based acceleration methods with CFG is analyzed very thoroughly.

2. Qualitative results demonstrate notable improvements in visual details and the spatio-temporal consistency of the generated frames.

**Weaknesses:**

1. Slightly missing justifications for the design choices of the proposed method. For instance, is the linear interpolation strategy sufficient for dynamically building the missing features? Are there any other options that the authors have considered? Using the proposed Dynamic Feature Reuse, how similar are the computed features compared to the existing caching methods? (Please guide me if I missed the descriptions)

2. Limited video-specific contributions. The proposed methodologies seem to applicable to any image diffusion model, of which comparison to previous works is missing, and I could not find any video-related novelties proposed. The notation $t = \{1 ... T\}$ in Equations seem to denote the diffusion sampling steps, and I think there is no notation for the temporal axis? Please correct me if I misunderstood any equations.

3. (Minor) Figure captions are not very descriptive. I would suggest to include the main points (Fig. 2, 8) or where to focus on (qualitative figures).

**Questions:**

1. For CFG-Cache, should we interpret the feature differences in the frequency domain in a similar way as MSE?

2. In Figure 7b, it seems to me that low-frequency components do not gradually shift into high-freq as the authors mention in L300-301, since the high-freq components differ both at the beginning and near the end. What kind of insights can we gain from this analysis?

3. In the experiments, PAB seems to significantly outperform $\Delta-DiT$, but there are no PAB results for CogVideoX and Vchitect. Also, why is the performance of $\Delta-DiT$ so low? Is it because of the inherent randomness of image diffusion models?

---

> ### Author Response · Authors · 2024-11-20
> **Response to Reviewer 4sX2 (part 1)**
>
> Thank you for your review! We address the concerns below:
>
>
> ***W1-Q1: Slightly missing justifications for the design choices of the proposed method. For instance, is the linear interpolation strategy sufficient for dynamically building the missing features? Are there any other options that the authors have considered?***
>
> Thank you for your suggestions.
>
> **(a)** We have included additional experiments in Appendix A.3 and A.5 to discuss the justifications behind the design choices and hyperparameters.
>
> **(b)** In Appendix A.3.1, we provide an additional discussion on the introduction of the feature bias term $F_{cache}^t - F_{cache}^{t+2}$ in Eq. (5), which aims to estimate the error between $F_{t-1}$ and $F^t_{cache}$. To achieve dynamic feature reuse, we also implement different strategies by controlling the $w(t)$ function in Eq. (5). In Appendix A.3.1, we add new experiments to discuss three designs: Constant weights $w(t)$, Learnable weights $w(t)$, and Linearly increasing $w(t)$. Table 5 (as illustrated below) in Appendix A.3.1 shows that Learnable weights performed best, with Linearly increasing achieving comparable results. Additionally, Fig. 13 (a) in Appendix A.3.1 demonstrates that the Learnable weights, obtained through optimization, gradually increase as sampling progresses. Considering effectiveness and simplicity, we chose Linearly increasing $w(t)$ as the default strategy. More details and discussion can be found in Appendix A.3.1.
>
>
> **(c)** As highlighted in our paper, Dynamic Feature Reuse makes cache-based methods feasible for real-world applications. We aimed to provide a straightforward yet effective approach and believe there is significant potential for further research on this topic. We will explore even better strategies for dynamic feature reuse in future work.
>
> | Method                     | LPIPS  | PSNR  | SSIM   |
> |----------------------------|--------|-------|--------|
> | Vanilla FR                | 0.0657 | 28.20 | 0.8785 |
> | Dynamic FR (w(t)=0.5)     | 0.0615 | 28.33 | 0.8889 |
> | Dynamic FR (learned w(t)) | 0.0596 | **28.45** | **0.8941** |
> | Dynamic FR (linear w(t))  | **0.0590** | 28.41 | 0.8938 |
>
> ***W1-Q2: Using the proposed Dynamic Feature Reuse, how similar are the computed features compared to the existing caching methods?***
>
> **(a) MSE similarity.** In the Fig. 10 (a), we compare the MSE between the target features (obtained from the original sampling process) and the features computed using existing caching methods (i.e., Vanilla Feature Reuse), as well as the MSE between the target features and those computed using our Dynamic Feature Reuse strategy. It can be observed that the **features computed with the Dynamic Feature Reuse strategy are closer to the target features.**
>
> **(b) Visual similarity.** Also, we provide more discussion and visualization about the comparison between Vanilla Feature Reuse (exisiting caching methods) and Dynamic Feature Reuse in Appendix A.3.1 and Fig. 13 (b). It can be observed that, compared to Vanilla Feature Reuse, Dynamic Feature Reuse obtains features that are closer to the original ones, with reduced residual intensity and a smaller range. The visual results also validate the effectiveness of our proposed Dynamic Feature Reuse in preserving fine details.
>
>
> ***W2: Limited video-specific contributions.***
>
> Thank you very much for your insight. We will emphasize the video-specific aspects for video generation in the paper.
>
>
> **(a)** For the proposed insights in Section 2.2 and 2.3, these analyses are based on the sequence features of videos, revealing the impact of current reuse methods on details synthesis of videos.
>
> **(b)** We did not specifically mention video frame information in our proposed method because of **the significant evolution of video generation models**. Early video diffusion models, such as the spatiotemporal decoupling framework (e.g., The Open-Sora, Open-Sora-Plan, and Latte), separated temporal and spatial modeling via **dedicated temporal and spatial attention**. Our method, combined with their temporal attention, is temporal-dynamic feature reuse, which is video-specific. Recent video diffusion models, such as CogVideoX, Vchitect 2.0, and Mochi, use a **fully self-attention layer** to learn the entire token sequence of the video, without distinguishing between spatial and temporal components, which is increasingly homogeneous with that of image synthesis models. Therefore, **to emphasize the generality of FasterCache across these methods**, we did not overly emphasize video-specific descriptions in the paper.
>
> **(c)** Our method can be adapted to image diffusion models. In Fig. 12, we show its potential on PixArt-Sigma. Compared to the state-of-the-art cache-based image synthesis acceleration method, $\Delta -$DiT, our approach maintains higher fidelity at the same acceleration level (e.g. 1.5× latency speedup, with **FasterCache achieving an SSIM score of 0.84 and $\Delta -$DiT an SSIM score of 0.80**).

---

> ### Author Response · Authors · 2024-11-20
> **Response to Reviewer 4sX2 (part 2)**
>
> ***W3: Figure captions are not very descriptive.***
>
> Thank you for your suggestion. We have revised and supplemented the captions in Fig. 2, Fig. 8, and the qualitative Fig. 9 as per your suggestion.
>
>
>
> ***Q1: For CFG-Cache, should we interpret the feature differences in the frequency domain in a similar way as MSE?***
>
> Thank you for your valuable insights.
>
> * In Fig. 7(b), we visualize the different trends of CFG features in the frequency domain. It can be observed that in the early stages after CFG-Cache starts to take effect, there is a significant difference between the conditional and unconditional features in both high-frequency and low-frequency components, with a larger difference in the low-frequency component. This is because, during the early steps of denoising in diffusion models, the focus is mainly on generating perception-related (e.g. layout and shape) low-frequency features along with a small amount of high-frequency information.
>
> * In the later steps of denoising, the low-frequency difference gradually decreases while the high-frequency difference increases. This is because, in these steps, the layout (low-frequency information) has mostly stabilized, and the focus shifts more towards generating details (high-frequency information). Thus, the difference gradually becomes dominated by high-frequency features.
>
> * Fig. 14 provides a more intuitive illustration of this process. To address this, we propose CFG-cache, which dynamically enhances both high-frequency and low-frequency components before reuse.
>
> ***Q2: Supplementary explanation for Figure 7 (b) and the insights derived from it.***
>
> Thank you for your insights. We have revised and supplemented Figure 7 (b) and the related explaination in Section 2.4 (L297- L304) to avoid potential misunderstandings.
>
> **(a)** Considering that in the early steps, there is a significant difference between the conditional and unconditional outputs, including both low-frequency and high-frequency parts, CFG-Cache reuse is applied after the initial few steps to maintain stable generation during the denoising process.
>
> **(b)** Lines 300-301 describe the phenomenon from the timesteps when CFG-Cache starts to take effects: "the conditional and unconditional outputs at the same timestep exhibit a significant bias in the low-frequency components, which progressively shifts to the high-frequency components in the later steps." This occurs because, as sampling process progresses, the layout (low-frequency information) has mostly stabilized, and the focus shifts towards generating details (high-frequency information). This process is also intuitive illustrated in Fig. 14 in Appendix A.3.2.
>
> ***Q3: PAB results for CogVideoX and Vchitect.***
>
> Before submitting FasterCache, the official PAB code did not support CogVideoX and Vchitect 2.0. Currently, PAB has added support for these two video models, and we have tested and supplemented the PAB results on CogVideoX and Vchitect 2.0 based on the code provided by PAB. The results are shown in the table below and have been updated in the paper in Table 1.
>
>
> | Method                         | MACs            | Speedup                  | Latency | VBench   | LPIPS   | SSIM    | PSNR    |
> |--------------------------------|---------|----------|---------|---------|---------|---------|---------|
> | CogVideoX (48 frames, 480P, T=50)     | 6.03 | 1×  | 78.48   | 80.18%   | -       | -       | -       |
> | CogVideoX + PAB               | 4.45           | 1.35×         | 57.98   | 79.76%   | 0.0860  | 0.8978  | 28.04   |
> | CogVideoX + FasterCache      | **3.71**   | **1.62×** | **48.44**   | **79.83%**   | **0.0766**  | **0.9066**  | **28.93**   |
> | Vchitect 2.0(40 frames, 480P, T=100)     | 14.57 | 1×  | 260.32  | 80.80%   | -       | -       | -       |
> | Vchitect 2.0 + PAB                | 12.20           | 1.26×          | 206.23  | 79.56%   | 0.0489  | 0.8806  | 27.38 |
> | Vchitect 2.0 + FasterCache        | **8.67**    | **1.67×**  | **156.13**  | **80.84%**   | **0.0282**  | **0.9224**  | **31.45**   |

---

> ### Author Response · Authors · 2024-11-20
> **Response to Reviewer 4sX2 (part 3)**
>
> ***Q4: Why is the performance of $\Delta-$DiT so low?***
>
>
> (a) $\Delta-$DiT is initially designed for image synthesis acceleration and does not have open-sourced code, so we implemented the $\Delta-$DiT method ourselves. During implementation, we strictly followed the description in the paper and experimented with various parameter configurations to ensure a fair comparison. In the PAB [r1] and adaptive caching [r2] papers, their conclusion also indicated that the performance of $\Delta-$DiT is inferior to PAB.
>
> (b) We think this performance difference primarily stems from inherent different feature interactions between images and videos. For video generation, token spatiotemporal interactions are more complex in the early stages of sampling. The use of $\Delta-$Cache on back blocks during the outline generation stage (early sampling stage) may introduce unavoidable perturbations, significantly impacting performance.
>
>
> [r1] Zhao, Xuanlei, et al. "Real-time video generation with pyramid attention broadcast." arXiv preprint arXiv:2408.12588(2024).
>
>
> [r2] Kahatapitiya, Kumara, et al. "Adaptive Caching for Faster Video Generation with Diffusion Transformers."arXiv preprint arXiv:2411.02397(2024).
>
>
> ---
> We deeply appreciate the reviewer's efforts in helping make our paper more clear and stronger.
>
> **If there are any further questions, we are happy to dicuss them!**

---

> > ### Author Response · Authors · 2024-11-23
> > **Follow-up**
> >
> > We hope our response has addressed your concerns. If there’s anything else you’d like us to clarify or discuss further, we’d greatly appreciate your feedback!
> >
> > Thank you for your time and consideration!

---

> ### Author Response · Authors · 2024-11-24
> **Additional Statistical Justifications for Dynamic Feature Reuse**
>
> We have included additional statistical experimental results in Appendix A.3.1 and Figure 13 to  provide further justification for the design of Dynamic Feature Reuse. We hope this better addresses your concern.
>
> If you have any further questions, we would be happy to discuss them with you.
>
> Thank you very much!

---

> ### Author Response · Authors · 2024-12-03
> **Kind Reminder**
>
> We sincerely appreciate the time and effort the reviewer has devoted to reviewing our submission. We have submitted our response and would like to follow up to ensure that it has adequately addressed the concerns.
>
> If you have any remaining questions or require further clarification, please feel free to let us know. We are keen to address them promptly before the discussion deadline and look forward to your feedback.
>
> Thank you once again for your valuable insights and comments!

---

### Author Response · Authors · 2024-11-20
**Global response and summary of changes in the revision**

We sincerely thank all reviewers for their time and effort in reviewing our paper!


In this work, we identify a common issue in previous cache-based methods: while features at adjacent timesteps are similar, ignoring the subtle differences and simply reusing them results in loss of detail. To address this, we propose the **Dynamic Feature Reuse** strategy, which introduces an additional feature bias term to preserve quality during acceleration. Additionally, through a comprehensive analysis of the video diffusion framework, we uncover significant redundancy in the unconditional computations of CFG, often overlooked by prior training-free acceleration methods. We introduce **CFG-Cache** to reduce this redundancy. By combining Dynamic Feature Reuse with CFG-Cache, FasterCache achieves significant acceleration while maintaining high fidelity. For example, on Vchitect 2.0, FasterCache achieved **80.84% VBench score and 1.67$\times$ speedup**, significantly outperforming the second-best result of **79.56% VBench score and 1.26$\times$ speedup**.


Below, we summarize the changes made in the revised paper:


1. We added the additional experiments about the justifications behind the hyperparameters or design choices in Appendix A.3.1 and A.5. We have also added supplementary explanations in the corresponding sections of the main text (L287 - L288). (Reviewer: 4sX2, 4a6E, zNbc)
2. We added Fig. 13 (b) to visualize the similarity of features computed by Dynamic Feature Reuse and existing caching based method (Vanilla Feature Reuse). (Reviewer: 4sX2)
3. We revised and supplemented the captions in Fig. 2, Fig. 8 and Fig. 9 for clarity. (Reviewer: 4sX2)
4. We have revised and supplemented Figure 7 (b) and the related explaination in Section 2.4 (L297 - L304). We also added a visual varations of CFG output in Fig. 14 and Appendix A.3.2. (Reviewer: 4sX2, zNbc)
5. We added the quantitative results of PAB on CogVideoX and Vchitect 2.0 in Table 1. (Reviewer: 4sX2)
6. We added the visual results of FasterCache based on larger scale video diffusion model (CogVideoX-5B and Mochi-10B) in Fig. 21 in Appendix A.4. (Reviewer: 4a6E)
7. We added a theoretical discussion on the effectiveness of Dynamic Feature Reuse and CFG-Cache in Appendix A.3.1 and Appendix A.3.2.  (Reviewer: 4a6E)
8. We added an experiment on the settings of CFG scale and negative prompt in Fig. 15 in Appendix A.3.3. (Reviewer: 4a6E)


The revised content is highlighted in blue.

We sincerely thank all reviewers again for their valuable suggestions, which have greatly helped strengthen our paper.


If you have any further questions, we would be happy to discuss them!

---

### Author Response · Authors · 2024-12-02
**Follow-up on Reviewers Feedback**

We sincerely thank all reviewers for their constructive comments and the time and effort they dedicated to evaluating our paper. We hope we have adequately addressed all concerns and respectfully request the reviewer to take this into account in the final evaluation.

Thank you very much!

---

### Meta-Review · Area_Chair_1hUU · 2024-12-15

**Metareview:**

Summary: This work proposes a dynamic feature-caching mechanism for inference-time video diffusion sampling. Unlike straightforward re-use of adjacent-time features, the method interpolates features of neighboring time steps in a weighted manner. Additionally, the authors identify and address a redundancy in classifier-free guidance (CFG), where conditional and unconditional features at the same time step overlap. They propose reusing conditional outputs to compute unconditional outputs, reducing redundancy. Experimental results demonstrate notable improvements in image quality and runtime performance.
Strengths: The main contribution is well-motivated, clearly illustrated, and supported by both qualitative and quantitative analyses. The work is highly relevant for accelerating video sampling, a process often hindered by computational expense.
Weaknesses: The contribution is somewhat narrow, focusing specifically on image/video diffusion sampling. The paper lacks a comparison with distillation-based methods that train smaller networks for similar acceleration goals.
Acceptance Reasoning:The proposed ideas, while simple, are highly relevant for speeding up diffusion sampling processes. The authors have sufficiently addressed most weaknesses raised by reviewers, leading to rating increases from some. Although not all reviewers responded after the rebuttal, the authors provided comprehensive responses to the concerns, addressing them effectively.

**Additional Comments On Reviewer Discussion:**

Reviewer 4sX2 appreciated the contributions but raised concerns about the choice of the linear interpolation strategy. In the rebuttal, the authors provided detailed experimental results supporting their linear and weighted techniques, addressing the reviewer’s concerns. However, the reviewer did not respond despite multiple follow-ups. Similarly, Reviewer 4a6E questioned the justification for the linear interpolation approach. The authors' rebuttal adequately addressed this concern, which the reviewer acknowledged.

---

### Decision · Program_Chairs · 2025-01-22

Accept (Poster)